# CLIPure: Purification in Latent Space via CLIP for Adversarially Robust Zero-Shot Classification

**Mingkun Zhang**[1,3], **Keping Bi**[2,3,*], **Wei Chen**[1,3,*], **Jiafeng Guo**[2,3], **Xueqi Cheng**[1,3]

[1]State Key Lab of AI Safety, CAS Key Lab of AI Safety,
  Institute of Computing Technology, Chinese Academy of Sciences, Beijing, China
[2]CAS Key Laboratory of Network Data Science and Technology,
  Institute of Computing Technology, Chinese Academy of Sciences, Beijing, China
[3]University of Chinese Academy of Sciences, Beijing, China
{zhangmingkun20z, bikeping, chenwei2022, guojiafeng, cxq}@ict.ac.cn

## Abstract

In this paper, we aim to build an adversarially robust zero-shot image classifier. We ground our work on CLIP, a vision-language pre-trained encoder model that can perform zero-shot classification by matching an image with text prompts "a photo of a <class-name>.". Purification is the path we choose since it does not require adversarial training on specific attack types and thus can cope with any foreseen attacks. We then formulate purification risk as the KL divergence between the joint distributions of the purification process of denoising the adversarial samples and the attack process of adding perturbations to benign samples, through bidirectional Stochastic Differential Equations (SDEs). The final derived results inspire us to explore purification in the multi-modal latent space of CLIP. We propose two variants for our CLIPure approach: *CLIPure-Diff* which models the likelihood of images' latent vectors with the DiffusionPrior module in DaLLE-2 (modeling the generation process of CLIP's latent vectors), and *CLIPure-Cos* which models the likelihood with the cosine similarity between the embeddings of an image and "a photo of a.". As far as we know, CLIPure is the first purification method in multi-modal latent space and CLIPure-Cos is the first purification method that is not based on generative models, which substantially improves defense efficiency. We conducted extensive experiments on CIFAR-10, ImageNet, and 13 datasets that previous CLIP-based defense methods used for evaluating zero-shot classification robustness. Results show that CLIPure boosts the SOTA robustness by a large margin, e.g., from 71.7% to **91.1%** on CIFAR10, from 59.6% to **72.6%** on ImageNet, and **108%** relative improvements of average robustness on the 13 datasets over previous SOTA. The code is available at https://github.com/TMLResearchGroup-CAS/CLIPure.

## 1 Introduction

Image classifiers are usually trained in a supervised manner with training data and evaluated on the corresponding test data until recently several vision-language models have emerged as zero-shot classifiers (Li et al., 2023; Radford et al., 2021; Li et al., 2022). Among them, CLIP (Radford et al., 2021) is an example that is popular, effective, and efficient. CLIP performs zero-shot classification by forming text prompts "a photo of a <class-name>." of all the candidate categories, and selecting the class with the highest similarity with the image embedding. Despite its efficacy, when facing adversarial attacks, its accuracy can drop to zero, similarly vulnerable to other neural classifiers.

Existing methods to enhance adversarial robustness follow two primary paths: adversarial training and purification. Adversarial Training (AT) (Madry et al., 2017; Rebuffi et al., 2021; Wang et al., 2023) incorporates adversarial examples into model training to boost robustness. It often achieves

---

*corresponding authors

advanced performance in defending against the same attacks while failing to defend against unseen attacks (Chen et al., 2023). FARE (Schlarmann et al., 2024) and TeCoA (Mao et al., 2022) are two AT approaches integrated with CLIP, which enhance CLIP's zero-shot classification robustness while harming clean accuracy significantly and do not generalize to other types of attacks. Adversarial purification (Song et al., 2017) seeks to eliminate adversarial perturbations by optimizing samples to align them with the distribution of benign samples. Instead of adversarial samples, this approach often requires a generative model that can model the probability of benign samples. It can handle unforeseen attacks but often performs worse than AT methods on seen attacks and has lower inference efficiency. In this paper, we aim to produce an adversarially robust zero-shot classifier, so we opt for integrating purification with CLIP. To explore a better purification method, we formalize the purification risk and theoretically

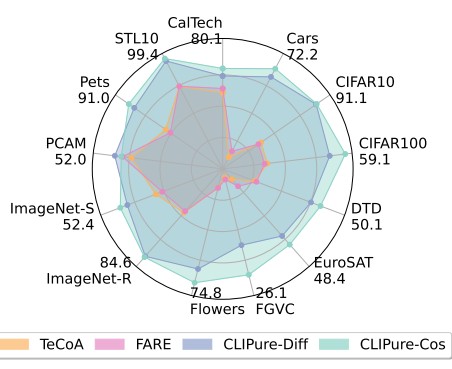

Figure 1: Adversarial robustness of two CLIPure versions versus adversarially trained CLIP models, evaluated against AutoAttack with $\ell_\infty = 4/255$ across 13 zero-shot classification datasets.

analyze what may affect purification performance. Concretely, inspired by Song et al. (2020) that models the diffusion process with bidirectional Stochastic Differential Equations (SDEs) (Anderson (1982)), we model the process of attack (i.e., adding perturbations to benign examples) with a forward SDE and purification (i.e., denoising adversarial examples) with a reverse SDE. Within this framework, we evaluate the risk of purification methods by measuring the KL divergence between the joint distributions of the purification and attack steps. After some derivation, we find that purification risk is related to 1) the negative KL divergence of the probability distributions of adversarial and benign examples (i.e., $-\mathrm{KL}(p(\boldsymbol{x}_{\mathrm{adv}})||p(\boldsymbol{x}_{\mathrm{ben}}))$), and 2) the $\ell_2$ norm of the gradients of adversarial samples' probability distribution (i.e., $\nabla \log p(\boldsymbol{x}_{\mathrm{adv}})$). This indicates that purification risk can be affected by: 1) the differences between $p(\boldsymbol{x}_{\mathrm{adv}})$ and $p(\boldsymbol{x}_{\mathrm{ben}})$; 2) the smoothness of $p(\boldsymbol{x}_{\mathrm{adv}})$ and possibly its dimension.

To the best of our knowledge, existing adversarial purification methods are all conducted in pixel space. Given the above factors that can affect purification risk, it is natural to ask: *are there purification approaches better than in pixel space?* As we know, pixel space is high-dimensional and sparse while latent embedding space is denser and smoother. Moreover, multi-modal latent representations are theoretically proven to have better quality than uni-modal (Huang et al., 2021). CLIP, as a vision-language-aligned encoder model, has shown the superiority of its multi-modal embeddings on many tasks (Radford et al., 2021). Accordingly, we propose to conduct purification in CLIP's latent space for adversarially robust zero-shot classification.

Our method - CLIPure has two variants that model the likelihood of images' latent vectors with CLIP differently: 1) CLIPure-Diff: a generative version that employs the DiffusionPrior module of DaLLE-2 (Ramesh et al., 2022) to model the likelihood of an image embedding with a diffusion model; 2) CLIPure-Cos: a discriminative version that models the likelihood with the cosine similarity between the image embedding and text embedding of a blank template "a photo of a .". Compared to likelihood modeling in pixel space and uni-modal latent space, we find that both our methods have several orders of magnitude larger KL divergence between the distributions of adversarial and benign samples than the former, and significantly higher than the latter (shown in Figure 2). Remarkably, CLIPure-Cos becomes the first purification method that does not rely on generative models and thus boosts defense efficiency by two or three orders of magnitude (See Table 3).

Since CLIP aligns image and text embeddings by their cosine similarities (Radford et al., 2021), vector lengths are not important to reflect the vector relationship in CLIP's latent space. Hence, during likelihood maximization in CLIPure, we normalize the latent vectors to unit vectors to diminish the effect of vector length. This is critical to our approach, as our experiments indicate that the purification process could be obstructed by vector magnitude and ultimately fail.

We compare the robustness of CLIPure and SOTA methods against the strongest adaptive attacks, AutoAttack(Croce & Hein, 2020), with different $\ell_2$ or $\ell_\infty$ bounds on various image classification tasks, including the popular CIFAR10, ImageNet, and 13 other datasets (e.g., CIFAR100, ImagetNet-R) that evaluate zero-shot classification robustness in FARE (Schlarmann et al., 2024)

and TeCoA (Mao et al., 2022). Note that CLIPure always conducts zero-shot classification and defense without the need for any dataset-specific training while the baselines can be any methods that are current SOTA. We are delighted to see that CLIPure boosts the SOTA robustness on all the datasets by a large margin, e.g., from 71.7% to 91.1% on CIFAR10 when $\ell_\infty = 8/255$, from 59.6% to 72.6% on ImageNet when $\ell_\infty = 4/255$. CLIPure achieves 45.9% and 108% relative improvements over previous SOTA - FARE(Schlarmann et al., 2024) regarding average robustness across the 13 zero-shot test datasets facing AutoAttack with $\ell_\infty = 2/255$ and $4/255$, depicted in Figure 1. Our work shows that purification in multi-modal latent space is promising for zero-shot adversarial robustness, shedding light on future research including but not limited to image classification.

## 2 RELATED WORK

**Zero-Shot Image Classification.** Unlike traditional models that are limited to predefined categories, vision-language models (VLMs) are trained on open-vocabulary data and align the embeddings of images and their captions into a common semantic space. This enables them to perform as zero-shot classifiers by matching the semantics of images to textual categories, offering superior generality and flexibility. CLIP (Radford et al., 2021), trained on extensive internet image-text pairs, achieves advanced results in zero-shot classification tasks. Additionally, other VLMs including Stable Diffusion (Rombach et al., 2022), Imagen (Saharia et al., 2022), and DaLLE-2 (Ramesh et al., 2022) also possess zero-shot classification capabilities (Li et al., 2023; Clark & Jaini, 2024).

**Adversarial Purification in Pixel Space.** A prevalent paradigm of adversarial purification aims to maximize the log-likelihood of samples to remove perturbations in pixel space. Since purification has no assumption of the attack type, enabling it to defend against unseen attacks using pre-trained generative models such as PixelCNN (Song et al., 2017), GANs (Samangouei, 2018), VAEs (Li & Ji, 2020), Energy-based models (Hill et al., 2020; Yoon et al., 2021), and Diffusion Models (Nie et al., 2022; Chen et al., 2023; Zhang et al., 2025). Owing to the capability of diffusion models, diffusion-based adversarial purification achieves state-of-the-art robustness among these techniques.

**CLIP-based Defense.** While CLIP achieves impressive accuracy in zero-shot classification, it remains vulnerable to imperceptible perturbations (Fort, 2021; Mao et al., 2022). Adversarially training the CLIP model on ImageNet / Tiny-ImageNet (Schlarmann et al., 2024; Mao et al., 2022; Wang et al., 2024) enhances its robustness but undermines its zero-shot capabilities and struggles against unseen attacks. Choi et al. (2025) suggests smoothing techniques for certification. Li et al. (2024a) advocates using robust prompts for image classification, but the defensive effectiveness is limited. Additionally, other research focuses on the out-of-distribution (OOD) robustness of the CLIP model (Tu et al., 2024; Galindo & Faria), which is orthogonal to our adversarial defense objectives.

## 3 PRELIMINARY: CLIP AS A ZERO-SHOT CLASSIFIER

In this section, we introduce how CLIP is trained and how CLIP acts as a zero-shot classifier. CLIP (Contrastive Language Image Pre-training) (Radford et al., 2021), consists of an image encoder $\text{Enc}^i$ and a text encoder $\text{Enc}^t$. It is trained on 400 million image-text pairs from the internet, aiming to align image embeddings with their corresponding text captions through contrastive learning:

$$\mathcal{L}_{\text{CLIP}} = -\frac{1}{2N} \sum_{n=1}^{N} \left[ \log \frac{\exp(\cos(\boldsymbol{z}_n^i, \boldsymbol{z}_n^t)/\tau)}{\sum_{m=1}^{N} \exp(\cos(\boldsymbol{z}_n^i, \boldsymbol{z}_m^t)/\tau)} + \log \frac{\exp(\cos(\boldsymbol{z}_n^i, \boldsymbol{z}_n^t)/\tau)}{\sum_{m=1}^{N} \exp(\cos(\boldsymbol{z}_m^i, \boldsymbol{z}_n^t)/\tau)} \right], \quad (1)$$

where $N$ represents the number of image-caption pairs, $\boldsymbol{z}_n^i = \text{Enc}^i(\text{image}_n)$ and $\boldsymbol{z}_n^t = \text{Enc}^t(\text{text}_n)$ are the embeddings of the $n$-th image and text respectively, $\tau$ is a temperature parameter, and $\cos(\cdot, \cdot)$ denotes the cosine similarity function.

This alignment enables CLIP to perform zero-shot classification by matching image embeddings with text embeddings of a template "a photo of a <class-name>.", where <class-name> iterates all the possible classes of a dataset. Without loss of generality, given an image, let $\boldsymbol{z}^i$ denote its CLIP-encoded image embedding, and $\boldsymbol{z}_c^t$ be the text embedding of a possible class description, i.e., $\boldsymbol{z}_c^t = \text{Enc}^t(\text{"a photo of a class } c\text{."})$. The predicted class $\hat{y}$ is determined by:

$$\hat{y} = \arg\max_c \cos(\boldsymbol{z}^i, \boldsymbol{z}_c^t). \quad (2)$$

For enhanced classification stability, as in Radford et al. (2021), we use 80 templates of diverse descriptions in combination with class names, such as "a *good* photo of <class-name>". In our experiments, each class $c$'s embedding, $z_c^t$ in Eq. 2 is the average text embedding of $c$ paired with all the templates.

# 4 CLIPURE: ADVERSARIAL PURIFICATION IN LATENT SPACE VIA CLIP

In this section, we outline the methodology of our CLIPure, focusing on adversarial purification within CLIP's latent space. We first define purification risk through a Stochastic Differential Equation (SDE) perspective and derive its lower bound in Section 4.1. Section 4.2 introduces the rationale for CLIPure to potentially achieve a smaller purification risk and two variants of modeling sample likelihood. In Section 4.3, we propose normalizing latent vectors to diminish the effect of vector length during purification to align with CLIP's latent space modeled using cosine similarity.

## 4.1 ADVERSARIAL PURIFICATION RISK

Considering that adversarial attacks progressively add perturbations to an image while purification gradually removing noise to restore the original image, we formulate both the attack and purification processes through the lens of Stochastic Differential Equations (SDEs). This framework allows us to propose a measure of purification risk based on the divergence between the attack and purification processes, providing insights into what affects purification effectiveness.

We formulate the attack process as a transformation from the benign distribution $p_{\text{ben}}(x)$ to an adversarial example distribution $p_{\text{adv}}(x)$ by an attack algorithm. Note that for simplicity we use $p(x)$ to represent $p_{\text{ben}}(x)$ in this paper. Take untargeted PGD-attack (Madry et al., 2017) for instance, the adversarial attack behavior on a benign sample $x_0$ can be described as:

$$\mathrm{d}x = \alpha \text{sign}(\nabla_x \mathcal{L}(\theta; x_t, y_{\text{true}}))\mathrm{d}t + \sigma \mathrm{d}w_t, \quad x_0 \sim p(x), \quad \text{s.t.,} \quad \|x_T - x_0\|_\rho \leq \epsilon, \quad (3)$$

where $\alpha$ represents the attack step size, $p(x)$ denotes the distribution of benign samples, $\mathcal{L}(\theta; x_t, y_{\text{true}})$ denotes the loss of $x_t$ classified by the model with parameters $\theta$ as the ground truth category $y_{\text{true}}$ at attack step $t$ (where $t \in [0, T]$), $\mathrm{d}w_t$ denotes the Wiener process (Brownian motion). The constant $\sigma$ serves as a scaling factor for the noise component and the adversarial example $x_T$ is bounded by $\epsilon$ in $\ell_\rho$ norm.

The corresponding reverse-time SDE (Anderson, 1982) of Eq. 3 describes the process from the adversarial example distribution $p_{\text{adv}}$ to the purified sample distribution $p_{\text{pure}}$, and is expressed as:

$$dx = [\alpha \underbrace{\text{sign}(\nabla_x \mathcal{L}(\theta; x_t, y_{\text{true}}))}_{\text{classifier guidance}} - \sigma^2 \underbrace{\nabla \log p(x_t)}_{\text{purification}}]\mathrm{d}t + \sigma \mathrm{d}\tilde{w}_t, \quad x_T \sim p_{\text{adv}}(x), \quad (4)$$

where $\log p(x_t)$ represents the log-likelihood of $x_t$ concerning the distribution of clean samples, analogous to the score function described in Score SDE (Song et al., 2020), $\mathrm{d}\tilde{w}_t$ represents the reverse-time Wiener process. A detailed discussion on the form of the reverse-time SDE can be found in Appendix A.

Note that in the reverse-time SDE, $t$ progresses from $T$ to $0$, implying that $\mathrm{d}t$ is negative. According to Eq. 4, the reverse SDE aims to increase the sample's log-likelihood while simultaneously decreasing the loss of classifying $x$ to $y_{\text{true}}$. In Eq. 4, the purification term is related to the common objective of adversarial purification $x_{\text{pure}} = \arg\max_x \log p(x)$. The classifier guidance term has been employed to enhance purification (Zhang et al., 2024), and their objective aligns well with Eq. 4. We will incorporate this guidance term with CLIPure in Appendix D.3 and see its impact.

Then, we define the joint distribution of the attack process described by the forward SDE in Eq. 3 as $\mathcal{P}_{0:T} = p(x_0 = x_{\text{ben}}, x_1, ..., x_T = x_{\text{adv}}) \in \mathbb{R}^{(T+1) \times d}$, where each $x_t \in \mathbb{R}^d$. For the purification process defined by the reverse-time SDE in Eq. 4, we denote the joint distribution as $\mathcal{Q}_{0:T} = p(x_0 = x_{\text{pure}}, x_1, ..., x_T = x_{\text{adv}})$. Here, $x_{\text{ben}}, x_{\text{adv}}$, and $x_{\text{pure}}$ denotes the benign sample, adversarial example, and purified sample respectively. We define the purification risk $\mathcal{R}(\mathcal{Q})$ by the KL divergence between the reverse SDE (corresponding to the purification process) and the joint distribution of forward SDE (representing the attack process):

$$\mathcal{R}(\mathcal{Q}) := \text{KL}(\mathcal{Q}_{0:T} \| \mathcal{P}_{0:T}) = \text{KL}(\mathcal{Q}_{0,T} \| \mathcal{P}_{0,T}) + \mathbb{E}_{\mathcal{Q}_{0,T}}[\text{KL}(\mathcal{Q}_{1:T-1|0,T} \| \mathcal{P}_{1:T-1|0,T})]$$
$$\geq \text{KL}(\mathcal{Q}_{0,T} \| \mathcal{P}_{0,T}). \quad (5)$$

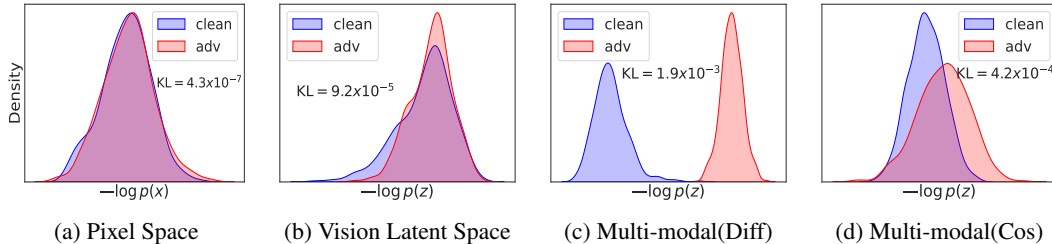

Figure 2: Negative log-likelihood estimated by diffusion models on (a) pixel space via EDM, (b) uni-modal latent space via VQVAE, (c) multi-modal latent space via DiffusionPrior, and (d) multi-modal latent space via CLIP (using cosine similarity for log-likelihood estimation). KL represents the value of $\mathrm{KL}(p(\boldsymbol{x}_{\mathrm{adv}})\|p(\boldsymbol{x}_{\mathrm{ben}}))$ discussed in Section 4.2 indicating the difference between clean and adversarial example distribution.

Then, we focus solely on the purified example, i.e., $\mathrm{KL}(\mathcal{Q}_{0,T}\|\mathcal{P}_{0,T})$ rather than the entire purification trajectory. The forward SDE in Eq. 3 describes the transformation from $p(\boldsymbol{x}_{\mathrm{ben}})$ to $p(\boldsymbol{x}_{\mathrm{adv}})$, enabling us to obtain the conditional probability $p(\boldsymbol{x}_{\mathrm{adv}}|\boldsymbol{x}_{\mathrm{ben}})$. Simultaneously, the reverse SDE in Eq. 4 supports the transformation from $p(\boldsymbol{x}_{\mathrm{adv}})$ back to $p(\boldsymbol{x}_{\mathrm{pure}})$, helping us to quantify $p(\boldsymbol{x}_{\mathrm{pure}}|\boldsymbol{x}_{\mathrm{adv}})$. Then we can derive that:

$$
\begin{aligned}
\mathcal{R}(\mathcal{Q}) \geq \mathrm{KL}(\mathcal{Q}_{0,T}\|\mathcal{P}_{0,T}) &= \mathrm{KL}(p(\boldsymbol{x}_{\mathrm{pure}}, \boldsymbol{x}_{\mathrm{adv}})\|p(\boldsymbol{x}_{\mathrm{ben}}, \boldsymbol{x}_{\mathrm{adv}})) \\
&= \frac{1}{2}\mathbb{E}_{\boldsymbol{x}_{\mathrm{adv}}}\left[\nabla \log p(\boldsymbol{x}_{\mathrm{adv}})^T \nabla \log p(\boldsymbol{x}_{\mathrm{adv}})\sigma^2 \Delta t\right] - \mathrm{KL}(p(\boldsymbol{x}_{\mathrm{adv}})\|p(\boldsymbol{x}_{\mathrm{ben}})),
\end{aligned}
\tag{6}
$$

where $\Delta t$ denotes a small time interval for attack and purification, related to the perturbation magnitude. A detailed proof of the result is provided in Appendix B.

The result in Eq. 6 highlights that the lower bound of the purification risk is influenced by two factors: 1) the smoothness of the log-likelihood function at adversarial examples and possibly the sample dimension, as indicated by the $\ell_2$ norm of $\nabla \log p(\boldsymbol{x}_{\mathrm{adv}})$, 2) the differences between the likelihood of clean and adversarial samples in the benign example space.

## 4.2 ADVERSARIAL PURIFICATION IN CLIP'S LATENT SPACE

In this section, we further explore how to achieve a smaller purification risk $\mathcal{R}(\mathcal{Q})$. Considering $\mathbb{E}_{\boldsymbol{x}_{\mathrm{adv}}}\left[\nabla \log p(\boldsymbol{x}_{\mathrm{adv}})^T \nabla \log p(\boldsymbol{x}_{\mathrm{adv}})\sigma^2 \Delta t\right]$ within $\mathcal{R}(\mathcal{Q})$ in Eq. 6, standard pixel space purification may lead to higher purification risks due to its sparsity and possibly peaked gradient distribution in high dimensionality. Thus, we are curious to investigate purification in latent space where the distribution of sample densities is more uniform and smoother.

$\mathrm{KL}(p(\boldsymbol{x}_{\mathrm{adv}})\|p(\boldsymbol{x}_{\mathrm{ben}}))$ in Eq. 6 implies representations that excel at detecting out-of-distribution adversarial examples are likely to carry a lower risk of purification errors. Huang et al. (2021) suggest that multi-modal latent spaces offer superior quality compared to uni-modal counterparts. Inspired by this observation, CLIP's well-aligned multi-modal latent space, where image embeddings are guided by the semantics of finer-grained words in an open vocabulary, may provide a foundation for purification with a lower risk.

To validate the efficacy of different spaces in modeling sample likelihood for adversarial purification, we focus on the term $\mathrm{KL}(p(\boldsymbol{x}_{\mathrm{adv}})\|p(\boldsymbol{x}_{\mathrm{ben}}))$ in the lower bound of the purification risk $\mathcal{R}(\mathcal{Q})$. As illustrated in Figure 2, we extract 512 samples from ImageNet (Deng et al., 2009) and generate adversarial examples by AutoAttack (Croce & Hein, 2020) on the CLIP (Radford et al., 2021) classifier. We explore four types of likelihood modeling approaches:

**In Pixel Space:** Sample likelihood of the joint distribution of image pixels is estimated by a generative model (we use an advanced diffusion model - EDM(Karras et al., 2022)). Figure 2a indicates that even EDM struggles to distinguish between clean and adversarial sample distributions at the pixel level. We use the Evidence Lower Bound (ELBO) to estimate log-likelihood, expressed as $\log p_\theta(\boldsymbol{x}) \geq -\mathbb{E}_{\boldsymbol{\epsilon},t}[\|\epsilon_\theta(\boldsymbol{x}_t, t) - \boldsymbol{\epsilon}\|_2^2] + C$, where $\boldsymbol{\epsilon} \sim \mathcal{N}(\mathbf{0}, \boldsymbol{I})$, and $C$ is typically negligible (Ho et al., 2020; Song et al., 2020).

---

**Algorithm 1 CLIPure**: Adversarial Purification in Latent Space via CLIP

---

**Required:** An off-the-shelf CLIP model including an image encoder $\text{Enc}^i$ and a text encoder $\text{Enc}^t$, textual embedding $\bar{z}^t$ of the blank templates (without class names, e.g., 'a photo of a.'), purification step $N$, step size $\eta$, and a DiffusionPrior model $\epsilon_\theta$ from DaLLE-2 (for generative version).
**Input:** Latent embedding of input example $z^i$
**Output:** label $y$
**for** $i = 1$ **to** $N$, **do**
  **step1: Obtain latent embedding in polar coordinates**
    direction $u = z^i / \|z^i\|_2^2$, magnitude $r = z^i / u$
  **step2: Estimate log-likelihood**

| **CLIPure-Diff** via DiffusionPrior $\epsilon_\theta$ | **CLIPure-Cos** via CLIP |
|---|---|
| sample $\epsilon \sim \mathcal{N}(0, 1)$ 
 $\log p(z^i) \leftarrow -\|\epsilon_\theta(z_t^i, t, \bar{z}^t) - \epsilon\|_2^2$ | $\log p(z^i) \leftarrow \cos(z^i, \bar{z}^t)$ |

  **step3: Update latent variable**
  $u \leftarrow u + \eta \frac{\partial \log p(z^i)}{\partial z^i} \cdot \frac{\partial z^i}{\partial u}$ , $z^i \leftarrow r \cdot u$
  **step4: Classification based on purified embedding**
    predict label $y$ across candidate categories according to Eq. 2
**end for**
return predicted label $y$

---

**In Vision Latent Space:** The likelihood of the joint distribution of an image embedding in a uni-modal space is estimated with the VQVAE component of the Stable Diffusion (SD) model (Rombach et al., 2022). Note that although SD is multi-modal, its VQVAE component is trained solely on image data and keeps frozen while training the other parameters, making it a vision-only generative model of latent vectors. Compared to Figure 2a, Figure 2b demonstrates improved capability in distinguishing clean and adversarial sample distributions.

**In Vision-Language Latent Space:** For CLIP's latent space, we present two approaches to estimate the log-likelihood of image embeddings, i.e., $\log p(z^i)$:

(1) **Diffusion-based** Likelihood Estimation (corresponds to *CLIPure-Diff*): The DiffusionPrior module in DaLLE-2 (based on CLIP) (Ramesh et al., 2022) models the generative process of image embeddings conditioned on text embeddings. Employing DiffusionPrior, we estimate the log-likelihood $\log p_\theta(z^i)$ by conditioning on a blank template "a photo of a .":

$$\log p_\theta(z^i) \approx \log p_\theta(z^i | \bar{z}^t) = -\mathbb{E}_{\epsilon,t}[|\epsilon_\theta(z_t^i, t, \bar{z}^t) - \epsilon|_2^2] + C, \tag{7}$$

where $\epsilon_\theta(z^i, t, z^t)$ is the UNet (Ronneberger et al., 2015) in DiffusionPrior (Ramesh et al., 2022), parameterized by $\theta$, with the noised image embedding $z_t^i$ at timestep $t$ under the condition of text embedding $z^t$, and $C$ is a constant typically considered negligible (Ho et al., 2020).

(2) **Cosine Similarity-based** Likelihood Estimation (corresponds to *CLIPure-Cos*): We estimate $\log p_\theta(z^i)$ by computing the cosine similarity between image embedding $z^i$ and the blank template's text embedding:

$$\log p_\theta(z^i) \propto \cos(z^i, \bar{z}^t). \tag{8}$$

Note that by modeling likelihood without using generative models, the defense efficiency can be significantly boosted. The inference time of CLIPure-Cos is only 1.14 times of the vanilla CLIP for zero-shot classification, shown in Table 3.

Figure 2c and Figure 2d show that CLIPure-Diff and CLIPure-Cos have several orders larger magnitude KL divergence between clean and adversarial samples than modeling likelihood in pixel space. Modeling likelihood in uni-modal latent space also leads to larger KL divergence than pixel space but is smaller than multi-modal space. It indicates that purification in CLIP's latent space is promising to have lower purification risk and enhance adversarial robustness.

### 4.3 ADVERSARIAL PURIFICATION BASED ON NORMALIZED UNIT VECTORS

Typically, purification in pixel space is conducted through gradient ascent on the sample $x$ by using the derivative of the log-likelihood $\log p(x)$: $x \leftarrow x + \alpha \nabla \log p(x)$, where $\alpha$ denotes the step size

for updates. However, given that the cosine similarities between vectors in CLIP's latent space are the criterion for alignment where vector lengths do not take effect, it is inappropriate to directly apply the typical purification update manner to this space. Thus, we normalize the image vectors to unit vectors at each purification step to diminish the effect of vector length. Specifically, we first normalize the vector $z^i = \text{Enc}^i(x)$, obtained from the CLIP model image encoder for an input sample $x$ (potentially an adversarial sample), to a unit vector $u = z^i/\|z^i\|_2^2$. Then we calculate the sample's log-likelihood $\log p(z^i)$ and compute the gradient $g_u$ by the chain rule:

$$g_u = \frac{\partial \log p(z^i)}{\partial u} = \frac{\partial \log p(z^i)}{\partial z^i} \cdot \frac{\partial z^i}{\partial u}. \tag{9}$$

This gradient $g_u$ is then used to update the direction $z^i$ for adversarial purification, detailed in Algorithm 1. Note that CLIPure is based on the original CLIP and does not need any extra training.

We also attempted adversarial purification by directly optimizing vectors instead of the normalized version. We experimented extensively with various steps, parameters, and momentum-based methods, but found it challenging to achieve robustness over 10% on ImageNet, indicating that it is difficult to find an effective purification path with vector lengths taking effect.

## 5 EXPERIMENTS

### 5.1 EXPERIMENTAL SETTINGS

**Datasets.** Following the RobustBench (Croce & Hein, 2020) settings, we assess robustness on CIFAR-10 and ImageNet. To compare against CLIP-based zero-shot classifiers with adversarial training (Schlarmann et al., 2024; Mao et al., 2022), we conduct additional tests across 13 image classification datasets (detailed in Appendix C.2). In line with Schlarmann et al. (2024), we randomly sampled 1000 examples from the test set for our evaluations.

**Baselines.** We evaluate the performance of **pixel space purification** strategies employing generative models, including Purify-EBM (Hill et al., 2020) and ADP (Yoon et al., 2021) based on Energy-Based Models; DiffPure based on Score SDE (Nie et al., 2022) and DiffPure-DaLLE2.Decoder based on Decoder of DaLLE2 (Ramesh et al., 2022); GDMP based on DDPM (Ho et al., 2020); and likelihood maximization approaches such as LM-EDM (Chen et al., 2023) based on the EDM model (Karras et al., 2022) and LM-DaLLE2.Decoder which adapts LM to the Decoder of DaLLE-2. Furthermore, we perform an ablation study adapting LM to the latent diffusion model to achieve **uni-modal latent space purification** using the Stable Diffusion Model (Rombach et al., 2022), denoted as LM-StableDiffusion. Furthermore, we also compare with the state-of-the-art **adversarial training** methods such as AT-ConvNeXt-L (Singh et al., 2024) and AT-Swin-L (Liu et al., 2024), along with innovative training approaches such as MixedNUTS (Bai et al., 2024) and MeanSquare (Amini et al., 2024), and methods that utilize DDPM and EDM to generate adversarial samples for training dataset expansion: AT-DDPM (Rebuffi et al., 2021) and AT-EDM (Wang et al., 2023). Moreover, we also compare CLIPure with adversarially trained CLIP model according to TeCoA (Mao et al., 2022) and FARE (Schlarmann et al., 2024). Additionally, we also evaluate the performance of **classifiers without defense** strategies, including CLIP, WideResNet (WRN), and Stable Diffusion. Due to space constraints, we list the details of baselines in Section C.4 in Appendix C.

**Adversarial Attack.** Following the setup used by FARE (Schlarmann et al., 2024), we employ **AutoAttack**'s (Croce & Hein, 2020) strongest white-box APGD for both targeted and untargeted attacks across 100 iterations, focusing on an $\ell_\infty$ threat model (typically $\epsilon = 4/255$ or $\epsilon = 8/255$) as well as $\ell_2$ threat model (typically $\epsilon = 0.5$) for evaluation. We also evaluate CLIPure against the **BPDA** (short for Backward Pass Differentiable Approximation) with **EOT** (Expectation Over Transformation) (Hill et al., 2020) and latent-based attack (Rombach et al., 2022) detailed in Appendix C.3.

### 5.2 MAIN RESULTS

In this section, we compare CLIPure-Diff and CLIPure-Cos with SOTA methods and examine the model robustness from various perspectives.

Table 1: Comparison of performance against AutoAttack under $\ell_\infty$ ($\epsilon = 8/255$) and $\ell_2$ ($\epsilon = 0.5$) threat model on CIFAR-10 dataset, showcasing various defense methods including adversarial training and purification mechanisms. We highlight defenses that operate in pixel or latent space ("defense space"), and indicates the modal information used in defense strategies with "V" for Vision and "V-L" for Vision-Language multimodal representations. We use underlining to highlight the best robustness for baselines, and bold font to denote the state-of-the-art (SOTA) across all methods.

| | Method | Defense Space | Latent Modality | Clean Acc (%) | Robust Acc (%) $\ell_\infty = 8/255$ | $\ell_2 = 0.5$ |
|---|---|---|---|---|---|---|
| w/o Defense | WRN-28-10 (Zagoruyko, 2016) | - | V | 94.8 | 0.0 | 0.0 |
| | StableDiffusion (Li et al., 2023) | - | V | 87.8 | 0.0 | 38.8 |
| | CLIP (Radford et al., 2021) | - | V-L | 95.2 | 0.0 | 0.0 |
| Adv. Train | AT-DDPM-$\ell_2$ (Rebuffi et al., 2021) | Pixel | V | 93.2 | 49.4 | 81.1 |
| | AT-DDPM-$\ell_\infty$ (Rebuffi et al., 2021) | Pixel | V | 88.8 | 63.3 | 64.7 |
| | AT-EDM-$\ell_2$ (Wang et al., 2023) | Pixel | V | 95.9 | 53.3 | 84.8 |
| | AT-EDM-$\ell_\infty$ (Wang et al., 2023) | Pixel | V | 93.4 | 70.9 | 69.7 |
| Other | TETRA (Blau et al., 2023) | Pixel | V | 88.2 | 72.0 | 75.9 |
| | RDC (Chen et al., 2023) | Pixel | V | 89.9 | 75.7 | 82.0 |
| Purify | LM - StableDiffusion | Latent | V | 37.9 | 6.9 | 8.6 |
| | DiffPure (Nie et al., 2022) | Pixel | V | 90.1 | 71.3 | 80.6 |
| | LM - EDM (Chen et al., 2023) | Pixel | V | 87.9 | 71.7 | 75.0 |
| | **Our CLIPure - Diff** | Latent | V-L | 95.2 | **88.0** | **91.3** |
| | **Our CLIPure - Cos** | Latent | V-L | 95.6 | **91.1** | **91.9** |

**Discussion of CLIPure-Diff and CLIPure-Cos.** We compare the performance of CLIPure-Diff and CLIPure-Cos against AutoAttack across CIFAR-10, ImageNet, and 13 datasets in Tables 1, 2, and 7 respectively, as well as defense against BPDA+EOT and latent-based attack in Table 4 and 5 in Appendix D. Results indicate that both models have achieved new SOTA performance on all the datasets and CLIPure-Cos uniformly outperforms CLIPure-Diff in clean accuracy and robustness. It is probably because the DiffusionPrior component used in CLIPure-Diff models the generation process by adding noise to the original image embeddings encoded by CLIP without diminishing the effect of vector magnitude. Specifically, the noise is added as $z_t^i = \sqrt{\bar\alpha} z_0^i + \sqrt{1 - \bar\alpha}\epsilon, \epsilon \sim \mathcal{N}(\mathbf{0}, \mathbf{I})$. We expect that the performance will be boosted if the generation process also eliminates the effect of vector length. In contrast, CLIPure-Cos has no such issue and it models the likelihood with cosine similarities that are consistent with CLIP.

**Comparisons with Purification in Pixel Space.** Compared to the SOTA purification methods , DiffPure (Nie et al., 2022) and LM-EDM (Chen et al., 2023) (the likelihood maximization approach based on the advanced diffusion model EDM (Karras et al., 2022)), our CLIPure achieves better adversarial robustness (shown in Table 1 and 2) as well as superior inference efficiency (shown in Table 3). Table 1 shows that on CIFAR10 under the $\ell_\infty$ and $\ell_2$ threat models, our CLIPure-Cos showed improvements of 27.1% and 22.5% over LM-EDM and improvements of 27.8% and 14.0% over DiffPure. On the ImageNet dataset, shown in Table 2, CLIPure-Cos achieves a relative increment of 288.2% over LM-EDM and 63.5% over DiffPure. Additionally, the inference efficiency of our CLIPure is multiple orders of magnitude higher than DiffPure and LM-EDM (see Table 3).

Moreover, in Table 2, we also compared the pixel space purification based on the Decoder component of DaLLE-2 (Ramesh et al., 2022) (LM-DaLLE2.Decoder). Results show that the LM-DaLLE2.Decoder suffers from a drop in accuracy, possibly because the diffusion architecture ADM (Dhariwal & Nichol, 2021) it employs is slightly inferior to EDM in terms of generation quality.

**Comparisons with Purification in Uni-modal Latent Space.** We evaluate latent space purification using Stable Diffusion (Rombach et al., 2022) (i.e., LM-StableDiffusion) on the CIFAR-10 dataset, with results detailed in Table 1. As discussed in Section 4.2, Stable Diffusion (SD) models an uni-modal image latent space through its VQVAE (Rombach et al., 2022). Direct comparisons are challenging due to SD and CLIP utilizing different training data and fundamentally distinct backbones (diffusion model versus discriminative model). As noted in Li et al. (2023), SD, as

Table 2: Performance comparison of defense methods on ImageNet against AutoAttack with $\ell_\infty$ threat model ($\epsilon = 4/255$). Indicates whether the methods use ImageNet training set for training as zero-shot. "V" stands for Vision, and "V-L" for Vision-Language multimodal representations.

| | Method | Defense Space | Latent Modality | Zero -Shot | Clean Acc (%) | Robust Acc (%) |
|---|---|---|---|---|---|---|
| w/o Defense | WideResNet-50 (Zagoruyko, 2016) | - | V | ✗ | 76.5 | 0.0 |
| | CLIP (Radford et al., 2021) | - | V-L | ✓ | 74.9 | 0.0 |
| Adv. Train | FARE (Schlarmann et al., 2024) | Latent | V-L | ✗ | 70.4 | 33.3 |
| | TeCoA (Mao et al., 2022) | Latent | V-L | ✗ | 75.2 | 44.3 |
| | AT-ConvNeXt-L (Singh et al., 2024) | Pixel | V | ✗ | 77.0 | 57.7 |
| | AT-Swin-L (Liu et al., 2024) | Pixel | V | ✗ | 78.9 | 59.6 |
| Others | MixedNUTS (Bai et al., 2024) | Pixel | V | ✗ | 81.5 | 58.6 |
| | MeanSparse (Amini et al., 2024) | Pixel | V | ✗ | 78.0 | 59.6 |
| Purify | LM - DaLLE2.Decoder | Pixel | V-L | ✓ | 36.9 | 9.2 |
| | DiffPure - DaLLE2.Decoder | Pixel | V-L | ✓ | 31.2 | 9.0 |
| | LM - EDM (Chen et al., 2023) | Pixel | V | ✗ | 69.7 | 18.7 |
| | DiffPure (Nie et al., 2022) | Pixel | V | ✗ | 71.2 | 44.4 |
| | **Our CLIPure - Diff** | Latent | V-L | ✓ | 73.1 | **65.0** |
| | **Our CLIPure - Cos** | Latent | V-L | ✓ | 76.3 | **72.6** |

an generative model, is designed for generation rather than classification tasks, so its zero-shot classification performance is not good enough, which limits its potential for robustness.

**Comparisons with CLIP-based Baselines.** Figure 1 and Table 7 show the zero-shot adversarial robustness on 13 datasets, compared to CLIP-based baselines enhanced with adversarial training, i.e., FARE (Schlarmann et al., 2024) and TeCoA (Mao et al., 2022), CLIPure-Diff and CLIPure-Cos surpass their best-reported average robustness by significant margins, 39.4% and 45.9% when $\ell_\infty = 2/255$, 95.4% and 108% when the attacks are stronger with $\ell_\infty = 4/255$. FARE and TeCoA often have much lower clean accuracy than vanilla CLIP due to their adversarial training on ImageNet, which harms zero-shot performance on other datasets. Additionally, Table 2 shows that even on ImageNet against the attacks they have been trained with, CLIPure still outperforms them by a huge margin (65% and 72.6% versus 33.3% and 44.3%). The robustness of CLIPure against unseen attacks is much better than their performance on seen attacks, showing that we are on the right path of leveraging the power of pre-trained vision-language models.

**More Experiments and Analysis.** Due to space constraints, in the Appendix, we include a detailed case study, showcasing the visualization of image embeddings during the purification process using a diffusion model in Figure 6 in Appendix D.1. Figure 11a in Appendix D.3 illustrates the effects of combining our approach with adversarial training and pixel space purification methods, while Figure 11b displays the outcomes of integrating classifier guidance. Additionally, we employ T-SNE to visualize the distribution of image and text embeddings in CLIP's latent space in Figure 12a and analyze the impact of step size on performance in Figure 12b.

## 5.3 INFERENCE EFFICIENCY

We evaluate the inference efficiency of our CLIPure and baseline models by measuring the inference time of an averaged over 100 examples from the CIFAR-10 dataset on a single 4090 GPU. The results are displayed in Table 3. Our CLIPure-Cos innovatively uses a discriminative approach for adversarial sample purification, achieving inference times comparable to those of discriminative models. Traditional purification methods typically leverage a generative model like diffusion and significantly involve complex inference procedures. Such complexity limits their applicability in efficiency-sensitive tasks, such as autonomous driving.

## 5.4 CLIP VARIANTS

To investigate the influence of backbone models and their scale, we evaluate the performance of CLIPure-Cos across various versions of CLIP (Radford et al., 2021) and its variants, including

Table 3: Comparison of different methods in terms of average inference time over 100 samples on CIFAR10. "Relative Time" expresses each method's inference time as a multiple of the CLIP model's time for classification. Dis. denotes purification with a discriminative model, and Gen. denotes that with a generative model.

| Method | CLIPure-Cos | CLIPure-Diff | DiffPure | LM-EDM | CLIP |
|---|---|---|---|---|---|
| Gen. or Dis. | Dis. | Gen. | Gen. | Gen. | Dis. |
| Inference Time (s) | $4.1 \times 10^{-4}$ | 0.01 | 2.22 | 0.25 | $3.6 \times 10^{-4}$ |
| Relative Time | 1.14x | 27.78x | 6166.67x | 694.44x | 1x |

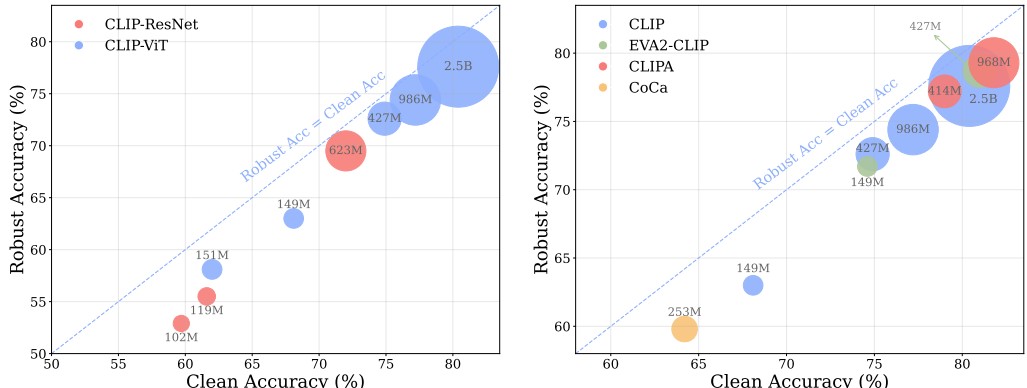

Figure 3: Accuracy and robustness (detailed in Table 6) of CLIPure-Cos for (Left) various versions of CLIP and (Right) different backbone models. The bubble size represents the number of parameters, which is also indicated alongside each bubble. The left figure presents CLIPure-Cos based on ResNet-based models (including RN50, RN101, RN50x64, marked in red) and ViT-based models (ViT-B-16, ViT-B-32, ViT-L-14, ViT-H-14, and ViT-bigG-14, marked in blue). The right figure depicts CLIPure-Cos based on CLIP (including ViT-B-16, ViT-L-14, ViT-H-14, ViT-bigG-14, marked in blue), EVA2-CLIP (including ViT-B-16 and ViT-L-14, marked in green), CLIPA (including ViT-L-14 and ViT-H-14, marked in red), and CoCa (ViT-B-32, marked in yellow). The blue dashed line represents the point where robust accuracy equals clean accuracy, serving as the upper bound of robustness, since successful defense against adversarial attacks hinges on accurate classification.

EVA2-CLIP (Sun et al., 2023), CLIPA (Li et al., 2024b), and CoCa (Yu et al.). The results in Table 6 and Figure 3 indicate: 1) Larger models generally exhibit stronger capabilities (i.e., clean accuracy), which in turn enhances the robustness of CLIPure. Notably, the CLIPure-Cos with CLIPA with ViT-H-14 achieves the best performance, with robustness reaching 79.3% on ImageNet—a truly remarkable achievement. 2) CLIPA-based CLIPure-Cos outperforms other CLIP models of comparable size in both accuracy and robustness. 3) ViT-based CLIPure-Cos models show better results than those based on ResNet.

## 6 CONCLUSION

We develop CLIPure, a novel adversarial purification method that operates within the CLIP model's latent space to enhance adversarial robustness on zero-shot classification without additional training. CLIPure consists of two variants: CLIPure-Diff and CLIPure-Cos, both achieving state-of-the-art performance across diverse datasets including CIFAR-10 and ImageNet. CLIPure-Cos, notably, does not rely on generative models, significantly enhancing defense efficiency. Our findings reveal that purification in a multi-modal latent space holds substantial promise for adversarially robust zero-shot classification, pointing the way for future research that extends beyond image classification.

ACKNOWLEDGMENTS

This work is supported by the Strategic Priority Research Program of the Chinese Academy of Sciences, Grant No. XDB0680101, CAS Project for Young Scientists in Basic Research under Grant No. YSBR-034, the National Key Research and Development Program of China under Grants No. 2023YFA1011602, the National Natural Science Foundation of China (NSFC) under Grants No. 62302486, the Innovation Project of ICT CAS under Grants No. E361140, the CAS Special Research Assistant Funding Project, the project under Grants No. JCKY2022130C039, the Strategic Priority Research Program of the CAS under Grants No. XDB0680102, and the NSFC Grant No. 62441229.

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

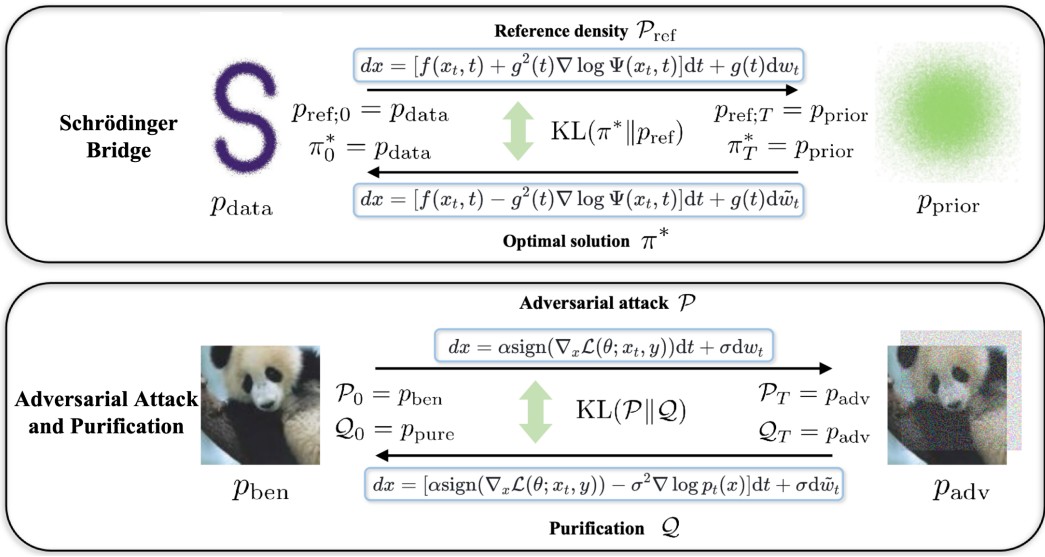

Figure 4: Comparison of the Schrödinger Bridge (Schrödinger, 1932) framework with our adversarial attack and purification process

## A    DISCUSSION OF THE REVERSE-TIME SDE

Motivated by the Schrödinger Bridge theory (Schrödinger, 1932; Tang et al., 2024) which models the transition between two arbitrary distributions, we utilize this theoretical framework to model the transformation between benign and adversarial sample distributions.

Within the Schrödinger Bridge framework, the forward Stochastic Differential Equation (SDE) describes a pre-defined transition from a data distribution $p_{\text{data}}$ to a prior distribution $p_{\text{prior}}$, noted as a reference density $p_{\text{ref}} \in \mathscr{P}_{T+1}$, the space of joint distributions on $\mathbb{R}^{(T+1) \times d}$, where $T$ represents the timesteps and $d$ denotes the data dimension. Consequently, we have $p_{\text{ref};0} = p_{\text{data}}$ and $p_{\text{ref};T} = p_{\text{prior}}$. Both $p_{\text{data}}$ and $p_{\text{prior}}$ are defined over the data space $\mathbb{R}^d$. The optimal solution $\pi^*$, starting from $p_{\text{prior}}$ and transferring to $p_{\text{data}}$, is termed the Schrödinger Bridge, described by the reverse SDE.

Specifically, the forward SDE is expressed as:

$$\mathrm{d}\boldsymbol{x} = [f(\boldsymbol{x}_t, t) + g^2(t)\nabla \log \Psi(\boldsymbol{x}_t, t)]\mathrm{d}t + g(t)\mathrm{d}\boldsymbol{w}_t, \quad x_0 \sim p_{\text{data}}, \tag{10}$$

where $f(\boldsymbol{x}_t, t)$ is the drift function and $g(t)$ is the diffusion term. $\boldsymbol{w}_t$ is the standard Wiener process, and $\Psi$ and $\tilde{\Psi}$ are time-varying energy potentials constrained by the following Partial Differential Equations (PDEs):

$$\begin{aligned} \frac{\partial \Psi}{\partial t} &= -\nabla_{\boldsymbol{x}} \Psi^T f - \frac{1}{2}\operatorname{Tr}(g^2 \nabla_{\boldsymbol{x}}^2 \Psi) \\ \frac{\partial \tilde{\Psi}}{\partial t} &= -\nabla_{\boldsymbol{x}} \tilde{\Psi}^T f - \frac{1}{2}\operatorname{Tr}(g^2 \nabla_{\boldsymbol{x}}^2 \tilde{\Psi}), \end{aligned} \tag{11}$$

such that $\Psi(\boldsymbol{x}, 0)\tilde{\Psi}(\boldsymbol{x}, 0) = p_{\text{data}}, \Psi(\boldsymbol{x}, T)\tilde{\Psi}(\boldsymbol{x}, T) = p_{\text{prior}}$.

Similarly, we define the mutual transformation between clean and adversarial sample distributions using coupled SDEs. The forward SDE describes the transformation process from benign to adversarial example distributions corresponding to an attacker's process. For an untargeted PGD attack, this is represented as:

$$\mathrm{d}\boldsymbol{x} = \alpha\operatorname{sign}(\nabla_{\boldsymbol{x}}\mathcal{L}(\theta; \boldsymbol{x}_t, y_{\text{true}}))\mathrm{d}t + \sigma\mathrm{d}\boldsymbol{w}_t, \quad \boldsymbol{x}_0 \sim p_{\text{ben}}, \tag{12}$$

where $\alpha$ represents the attack step size, $\mathcal{L}(\theta; \boldsymbol{x}_t, y_{\text{true}})$ denotes the loss when the adversarial example $\boldsymbol{x}_t$ is classified by the model with parameters $\theta$ as the ground truth category $y_{\text{true}}$ at attack step $t$ (where $t \in [0, T]$), $\mathrm{d}\boldsymbol{w}_t$ denotes the Wiener process (Brownian motion) to express more general cases, and $\sigma$ is a constant. The adversarial example $\boldsymbol{x}_T$ is bound by $\epsilon$ in $l_p$ norm, i.e., $\|\boldsymbol{x}_T - \boldsymbol{x}_0\|_p \leq \epsilon$.

According to Eq. 12, we obtain formulation of the drift function and diffusion term is

$$f(\boldsymbol{x}_t, t) = \alpha \mathrm{sign}(\nabla_{\boldsymbol{x}} \mathcal{L}(\theta; \boldsymbol{x}_t, y_{\mathrm{true}}))$$
$$g(t) = \sigma$$
(13)

According to (Anderson, 1982; Song et al., 2020), the corresponding reverse-time SDE is:

$$\mathrm{d}\boldsymbol{x} = [f(\boldsymbol{x}_t, t) - g^2(t)\nabla_{\boldsymbol{x}} \log p(\boldsymbol{x})]\mathrm{d}t + \sigma \mathrm{d}\tilde{\boldsymbol{w}}_t.$$
(14)

Thus, we can derive the corresponding reverse SDE of Eq. 12 as

$$dx = [\alpha \mathrm{sign}(\nabla_{\boldsymbol{x}} \mathcal{L}(\theta; \boldsymbol{x}_t, y_{\mathrm{true}})) - \sigma^2 \nabla \log p(\boldsymbol{x}_t)]\mathrm{d}t + \sigma \mathrm{d}\tilde{w}_t, \quad \boldsymbol{x}_T \sim p_{\mathrm{adv}},$$
(15)

where $\log p(\boldsymbol{x})$ is the log-likelihood of the marginal distribution of benign samples, also known as the score function in Score SDE (Song et al., 2020). $\mathrm{d}\tilde{w}_t$ denotes the reverse-time Wiener process as time flows from $t = T$ to $t = 0$.

Note that the coupled SDEs described in Eq. 12 and Eq. 15 are special cases of Eq. 11 when $\nabla \log \Psi(\boldsymbol{x}_t, t) = 0$ and $\nabla \log \tilde{\Psi}(\boldsymbol{x}_t, t) = \nabla \log p(\boldsymbol{x}_t)$.

Without loss of generality, other adversarial attack methods can typically be described using the forward SDE. Therefore, our proposed approach of modeling adversarial attacks and purification processes using coupled SDEs is universally applicable.

## B DETAILED PROOF OF PURIFICATION RISK FUNCTION

We first define the joint distribution of the attack process described by the forward SDE in Eq. 3 as $\mathcal{P}_{T+1}$, which spans $\mathbb{R}^{(T+1) \times d}$. Thus we have $\mathcal{P}_0 = p_{\mathrm{ben}}$ and $\mathcal{P}_T = p_{\mathrm{adv}}$. For the purification process, we designate the joint distribution as $\mathcal{Q}_{T+1}$, also defined over $\mathbb{R}^{(T+1) \times d}$, with $\mathcal{Q}_0 = p_{\mathrm{adv}}$ and $\mathcal{Q}_T = p_{\mathrm{pure}}$. $T$ represents the timestep, $p_{\mathrm{ben}}$, $p_{\mathrm{adv}}$ and $p_{\mathrm{pure}}$ denote the distribution of benign, adversarial, and purified samples.

According to the results of Léonard (2014), we have

$$\mathrm{KL}(\mathcal{Q}_{0:T}|\mathcal{P}_{0:T}) = \mathrm{KL}(\mathcal{Q}_{0,T}|\mathcal{P}_{0,T}) + \mathbb{E}_{\mathcal{Q}_{0,T}}[\mathrm{KL}(\mathcal{Q}_{1:T-1|0,T}|\mathcal{P}_{1:T-1|0,T})].$$
(16)

Since we focus on the differences between the purified and benign examples rather than the entire purification path, we concentrate on $\mathrm{KL}(\mathcal{Q}_{0,T}\|\mathcal{P}_{0,T})$. Thus, we continue to derive:

$$
\begin{aligned}
&\mathrm{KL}(\mathcal{Q}_{0,T}\|\mathcal{P}_{0,T}) \\
&= \mathrm{KL}(p(\boldsymbol{x}_{\mathrm{adv}}, \boldsymbol{x}_{\mathrm{pure}})|p_{\mathrm{attack}}(\boldsymbol{x}_{\mathrm{ben}}, \boldsymbol{x}_{\mathrm{adv}})) \\
&= \int \int p(\boldsymbol{x}_{\mathrm{adv}}) p(\boldsymbol{x}_{\mathrm{pure}}|\boldsymbol{x}_{\mathrm{adv}}) \log \frac{p(\boldsymbol{x}_{\mathrm{adv}}) p(\boldsymbol{x}_{\mathrm{pure}}|\boldsymbol{x}_{\mathrm{adv}})}{p(\boldsymbol{x}_{\mathrm{ben}}) p(\boldsymbol{x}_{\mathrm{adv}}|\boldsymbol{x}_{\mathrm{ben}})} \, d\boldsymbol{x}_{\mathrm{pure}} \, d\boldsymbol{x}_{\mathrm{adv}} \\
&= \int p(\boldsymbol{x}_{\mathrm{adv}}) \left( \int p(\boldsymbol{x}_{\mathrm{pure}}|\boldsymbol{x}_{\mathrm{adv}}) \log \frac{p(\boldsymbol{x}_{\mathrm{pure}}|\boldsymbol{x}_{\mathrm{adv}})}{p(\boldsymbol{x}_{\mathrm{adv}}|\boldsymbol{x}_{\mathrm{ben}})} \, d\boldsymbol{x}_{\mathrm{pure}} \right) d\boldsymbol{x}_{\mathrm{adv}} - \int p(\boldsymbol{x}_{\mathrm{adv}}) \log \frac{p(\boldsymbol{x}_{\mathrm{adv}})}{p(\boldsymbol{x}_{\mathrm{ben}})} \, d\boldsymbol{x}_{\mathrm{adv}} \\
&= \mathbb{E}_{\boldsymbol{x}_{\mathrm{adv}}} \left[ \mathrm{KL}(p(\boldsymbol{x}_{\mathrm{pure}}|\boldsymbol{x}_{\mathrm{adv}})\|p(\boldsymbol{x}_{\mathrm{adv}}|\boldsymbol{x}_{\mathrm{ben}})) \right] - \mathrm{KL}(p(\boldsymbol{x}_{\mathrm{adv}})\|p(\boldsymbol{x}_{\mathrm{ben}})),
\end{aligned}
$$
(17)

where $p(\boldsymbol{x}_{\mathrm{adv}}|\boldsymbol{x}_{\mathrm{ben}})$ represents the conditional probability transitioning from $\boldsymbol{x}_{\mathrm{ben}}$ to $\boldsymbol{x}_{\mathrm{adv}}$ as described by the forward SDE, and $q(\boldsymbol{x}_{\mathrm{pure}}|\boldsymbol{x}_{\mathrm{adv}})$ represents the conditional probability transitioning from $\boldsymbol{x}_{\mathrm{adv}}$ to $\boldsymbol{x}_{\mathrm{pure}}$ as described by the reverse SDE. This formulation underscores the effectiveness of the purification process by measuring how well the adversarial transformations are reversed.

Given that the perturbations added by attackers are typically imperceptible, over a small time interval $\Delta t$, the solution of the forward SDE described by Eq. 3 approximates a normal distribution:

$$p(\boldsymbol{x}_{\mathrm{adv}}|\boldsymbol{x}_{\mathrm{ben}}) \sim \mathcal{N}(\boldsymbol{x}_{\mathrm{ben}} + \alpha \, \mathrm{sign}(\nabla_x \mathcal{L})\Delta t, \sigma^2 \Delta t).$$
(18)

Similarly, the reverse SDE process described by Eq. 4 approximates a normal distribution:

$$p(\boldsymbol{x}_{\mathrm{pure}}|\boldsymbol{x}_{\mathrm{adv}}) \sim \mathcal{N}(\boldsymbol{x}_{\mathrm{adv}} - (\alpha \, \mathrm{sign}(\nabla_{\boldsymbol{x}_{\mathrm{adv}}} \mathcal{L}(\theta; \boldsymbol{x}_{\mathrm{adv}}, y_{\mathrm{true}}))) - \sigma^2 \nabla \log p(\boldsymbol{x}_{\mathrm{adv}}))\Delta t, \sigma^2 \Delta t).$$
(19)

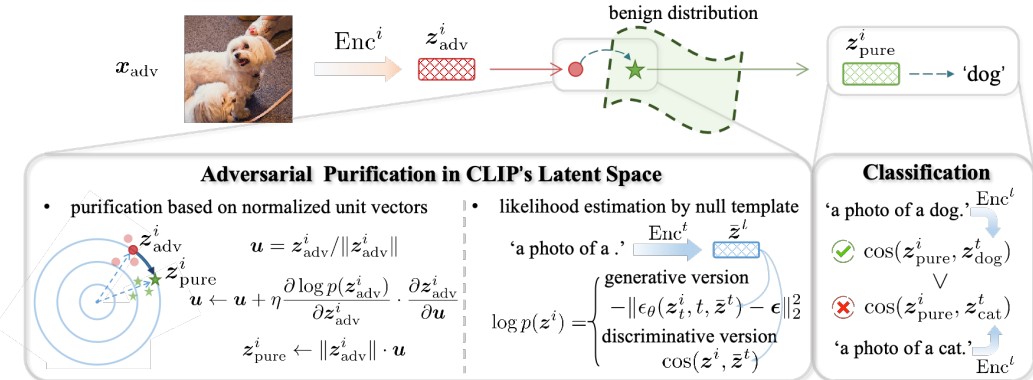

Figure 5: Illustration of the process of CLIPure including the purification in latent space and zero-shot classification, detailed in Algorithm 1.

The KL divergence formula for two normal distributions is:

$$KL(\mathcal{N}(\mu_0, \Sigma_0) \| \mathcal{N}(\mu_1, \Sigma_1)) = \frac{1}{2}\left(\text{tr}(\Sigma_1^{-1}\Sigma_0) + (\mu_1 - \mu_0)^T\Sigma_1^{-1}(\mu_1 - \mu_0) - k + \log\frac{\det\Sigma_1}{\det\Sigma_0}\right).$$
(20)

When $\Sigma_0 = \Sigma_1 = \sigma^2\Delta t$, the KL divergence reduced as

$$KL(\mathcal{N}(\mu_0, \Sigma_0) \| \mathcal{N}(\mu_1, \Sigma_1)) = \frac{1}{2\sigma^2\Delta t}\|\mu_1 - \mu_0\|^2.$$
(21)

Thus we have

$$\mathbb{E}_{\boldsymbol{x}_{\text{adv}}}\left[\text{KL}(p(\boldsymbol{x}_{\text{pure}}|\boldsymbol{x}_{\text{adv}})\|p(\boldsymbol{x}_{\text{adv}}|\boldsymbol{x}))\right]$$
$$= \mathbb{E}_{\boldsymbol{x}_{\text{adv}}}\left[\nabla\log p(\boldsymbol{x}_{\text{ben}})^T\nabla\log p(\boldsymbol{x}_{\text{ben}})\sigma^2\Delta t\right].$$
(22)

Thus we have proved the result in Eq. 6 that

$$\text{KL}(\mathcal{Q}_{0,T}|\mathcal{P}_{0,T}) = \text{KL}(p(\boldsymbol{x}_{\text{adv}}, \boldsymbol{x}_{\text{pure}})|p(\boldsymbol{x}, \boldsymbol{x}_{\text{adv}}))$$
$$= \mathbb{E}_{\boldsymbol{x}_{\text{adv}}}\left[\text{KL}(q(\boldsymbol{x}_{\text{pure}}|\boldsymbol{x}_{\text{adv}})\|p(\boldsymbol{x}_{\text{adv}}|\boldsymbol{x}_{\text{ben}}))\right] - \text{KL}(p(\boldsymbol{x}_{\text{adv}})\|p(\boldsymbol{x}))$$
$$= \frac{1}{2}\mathbb{E}_{\boldsymbol{x}_{\text{adv}}}\left[\nabla\log p(\boldsymbol{x}_{\text{adv}})^T\nabla\log p(\boldsymbol{x}_{\text{adv}})\sigma^2\Delta t\right] - \text{KL}(p(\boldsymbol{x}_{\text{adv}})\|p(\boldsymbol{x}_{\text{ben}})).$$
(23)

## C EXPERIMENTAL SETUP

### C.1 IMPLEMENTATION DETAIL

Following the implementation of pixel space purification (including Chen et al. (2023) with code and Zhang et al. (2023) with code ), CLIPure employs gradient ascent to maximize the log-likelihood of samples, which is formulated through: (1) cosine similarity (CLIPure-Cos) and (2) mean squared error (CLIPure-Diff). We conduct 10 purification steps on image embeddings obtained from CLIP's image encoder, with a step size of 30 for both CLIPure-Cos and CLIPure-Diff, to update directly on the vector direction of image embeddings. For CLIPure-Diff, we use the off-the-shelf DiffusionPrior model from DaLLE 2 (Ramesh et al., 2022). Experiments indicate that timesteps in the range of 900 to 1000 yield better defense outcomes when estimating likelihood via DaLLE2.Diffusion, hence we select this range for estimation.

For the blank template used in purification, we employ 80 diverse description templates combined with class names, such as "a *good* photo of <class-name>" to enhance stability, following zero-shot classification strategies. Consistently, each class $c$'s text embedding, $z_c^t$, as referred to in Eq. 2, is computed as the average text embedding combined with all templates.

Table 4: Performance comparison of defense methods on CIFAR-10 against BPDA with EOT-20 under $\ell_\infty$ ($\epsilon = 8/255$) norm bound. We use underlining to highlight the best robustness for baselines, and bold font to denote the state-of-the-art (SOTA) across all methods.

| Method | Clean Acc (%) | Robust Acc (%) |
|---|---|---|
| FARE (Schlarmann et al., 2024) | 77.7 | 8.5 |
| TeCoA (Mao et al., 2022) | 79.6 | 10.0 |
| Purify - EBM (Hill et al., 2020) | 84.1 | 54.9 |
| LM - EDM (Chen et al., 2023) | 83.2 | 69.7 |
| ADP (Yoon et al., 2021) | 86.1 | 70.0 |
| RDC (Chen et al., 2023) | 89.9 | 75.7 |
| GDMP (Wang et al., 2022) | 93.5 | 76.2 |
| DiffPure (Nie et al., 2022) | 90.1 | 81.4 |
| **Our CLIPure - Diff** | 95.2 | **94.6** |
| **Our CLIPure - Cos** | 95.6 | **93.5** |

## C.2 DATASETS FOR ZERO-SHOT PERFORMANCE

In order to evaluate the zero-shot performance of our CLIPure and adversarially trained CLIP model including FARE (Schlarmann et al., 2024) and TeCoA (Mao et al., 2022), we follow the settings and evaluation metrics used by FARE and outlined in the CLIP-benchmark[1]. We evaluate the clean accuracy and adversarial robustness across 13 datasets against an $\ell_\infty$ threat model with $\epsilon = 4/255$ and $\epsilon = 2/255$. The datasets for zero-shot performance evaluation include CalTech101 (Griffin et al., 2007), StanfordCars (Krause et al., 2013), CIFAR10, CIFAR100 (Krizhevsky et al., 2009), DTD (Cimpoi et al., 2014), EuroSAT (Helber et al., 2019), FGVC Aricrafts (Maji et al., 2013), Flowers (Nilsback & Zisserman, 2008), ImageNet-R (Hendrycks et al., 2021), ImageNet-S (Wang et al., 2019), PCAM (Veeling et al., 2018), OxfordPets (Parkhi et al., 2012), and STL10 (Coates et al., 2011).

## C.3 ADVERSARIAL ATTAKS

**BPDA with EOT.** To compare our approach with purification methods that do not support gradient propagation, we conducted evaluations using the BPDA (Backward Pass Differentiable Approximation) (Athalye et al., 2018) attack method, augmented with EOT=20 (Expectation over Transformation) to mitigate potential variability due to randomness. As depicted in Table 4, our CLIPure-Cos model significantly outperforms traditional pixel-space purification methods based on generative models such as EBM (Hill et al., 2020), DDPM (Chen et al., 2023), and Score SDE (Nie et al., 2022), despite being a discriminative model. Notably, our CLIPure demonstrates higher robust accuracy compared to the AutoAttack method under BPDA+EOT-20 attack. This is attributed to the fact that adaptive white-box attacks are precisely engineered to target the model, leading to a higher attack success rate.

**Latent-based Attack.** Shukla & Banerjee (2023) introduced a latent-based attack method, utilizing generative models such as GANs (Goodfellow et al., 2014) to modify the latent space representations and generate adversarial samples. We applied this approach to the CIFAR-10 dataset's test set, and the results, displayed in Table 5, reveal that unbounded attacks achieve a higher success rate than bounded attacks. Despite these aggressive attacks, our CLIPure method retains an advantage over baseline approaches. This is attributed to CLIPure leveraging the inherently rich and well-trained latent space of the CLIP model, which was trained on diverse and sufficient datasets, allowing it to effectively defend against unseen attacks without the need for additional training.

---

[1]Available at `https://github.com/LAION-AI/CLIP_benchmark/`

Table 5: Performance comparison of defense methods on CIFAR-10 against unbounded latent-based attack (Shukla & Banerjee, 2023) using a generative adversarial network (GAN) (Goodfellow et al., 2014).

| Method | Clean Acc (%) | Robust Acc (%) |
|---|---|---|
| ResNet-18 | 80.2 | 12.4 |
| FARE (Schlarmann et al., 2024) | 77.7 | 61.7 |
| TeCoA (Mao et al., 2022) | 79.6 | 63.7 |
| **Our CLIPure - Diff** | 95.2 | **69.2** |
| **Our CLIPure - Cos** | 95.6 | **70.8** |

## C.4 BASELINES

In this section, we discuss the detailed experimental settings of the baselines mentioned in Section 5.1. For the CIFAR-10 dataset results shown in Table 1, we use the off-the-shelf Stable Diffusion model as provided by Li et al. (2023), employing it as a zero-shot classifier. Our LM-StableDiffusion method adapts this model to a likelihood maximization approach. Other methods were applied directly according to the model in the original papers for CIFAR-10 without additional training.

For the ImageNet dataset experiments detailed in Table 2, FARE (Schlarmann et al., 2024) and TeCoA (Mao et al., 2022) involve testing checkpoints that were adversarially trained on the ImageNet dataset specifically for an $\ell_\infty$ threat model with $\epsilon = 4/255$. The LM-DaLLE2.Decoder method utilizes the Decoder module from the ViT-L-14 model of DaLLE2 (Ramesh et al., 2022) provided by OpenAI, adapted to the likelihood maximization method. DiffPure-DaLLE2.Decoder adapts the same Decoder module to the DiffPure (Nie et al., 2022) purification method. Other methods use models as provided and trained in the original papers.

Regarding the zero-shot performance across 13 datasets shown in Table 7, we opt for the adversarially trained models of TeCoA (Mao et al., 2022) and FARE (Schlarmann et al., 2024) with an $\ell_\infty$ threat model at $\epsilon = 4/255$, as they exhibited superior performance compared to those trained with $\epsilon = 2/255$.

In the results presented in Table 4 and Table 5, we conduct an adversarial evaluation using models as specified in the literature without any additional fine-tuning.

# D MORE EXPERIMENTAL RESULTS

## D.1 CASE STUDY

To visualize the purification path of CLIPure, we employ the Decoder of DaLLE2 (Ramesh et al., 2022), which models the generation process from image embeddings to images through a diffusion model. As shown in Figure 6, starting from an adversarial example initially classified as a 'stupa,' CLIPure modifies the semantic properties of the image embedding. By the second step, the image is purified to 'mountain,' closely aligning with the semantics of the original image, which also contains mountainous features. Subsequently, after multiple purification steps, the image embedding is accurately classified as the 'valley' category.

In Figure 10, we analyze the distribution of cosine similarity between the embeddings of clean images and adversarial examples with words to understand the semantics of image embeddings. In Figure 8, we also highlight the specific top-ranked categories for the benign and adversarial examples. We observe that the cosine similarity distribution for words close to the clean image is stable and resembles the prior distribution shown in Figure 9a. In contrast, the distribution of cosine similarity for the adversarial examples shows anomalies, with the top-ranked adversarial labels displaying abnormally high cosine similarities. This could potentially offer new insights into adversarial sample detection and defense strategies.

Table 6: Performance of CLIPure-Cos based on various versions of CLIP (Radford et al., 2021), EVA2-CLIP (Sun et al., 2023), CLIPA (Li et al., 2024b), and CoCa (Yu et al.) under the $\ell_\infty$ threat model ($\epsilon = 4/255$) on the ImageNet dataset. "Param" denotes the number of parameters. The prefix "RN" refers to ResNet-based (He et al., 2016) methods, while "ViT" indicates Vision Transformer-based (Alexey, 2020) approaches.

| Model | Version | Param (M) | w/o Defense | | CLIPure | |
|---|---|---|---|---|---|---|
| | | | Acc (%) | Rob (%) | Acc (%) | Rob (%) |
| | RN50 | 102 | 59.7 | 0.0 | 60.0 | 52.9 |
| | RN101 | 119 | 61.6 | 0.0 | 61.9 | 55.5 |
| | RN50x64 | 623 | 72.0 | 0.0 | 72.3 | 69.5 |
| CLIP | ViT-B-16 | 149 | 68.1 | 0.0 | 68.2 | 63.0 |
| (Radford et al., 2021) | ViT-B-32 | 151 | 62.0 | 0.0 | 62.0 | 58.1 |
| | ViT-L-14 | 427 | 74.9 | 0.0 | 76.3 | 72.6 |
| | ViT-H-14 | 986 | 77.2 | 0.0 | 77.4 | 74.4 |
| | ViT-bigG-14 | 2539 | 80.4 | 0.0 | 80.4 | 77.6 |
| EVA2-CLIP | ViT-B-16 | 149 | 74.6 | 0.0 | 74.7 | 71.7 |
| (Sun et al., 2023) | ViT-L-14 | 427 | 81.0 | 0.0 | 80.7 | 78.7 |
| CLIPA | ViT-L-14 | 414 | 79.0 | 0.0 | 79.0 | 77.2 |
| (Li et al., 2024b) | ViT-H-14 | 968 | 81.8 | 0.0 | **81.5** | **79.3** |
| CoCa (Yu et al.) | ViT-B-32 | 253 | 64.2 | 0.0 | 63.8 | 59.8 |

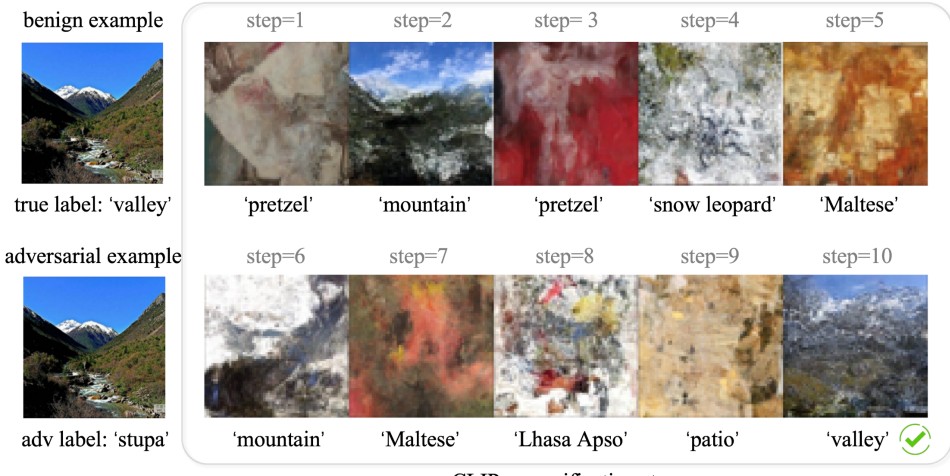

CLIPure purification steps

Figure 6: Purification process of our CLIPure model for an image of a 'valley' adversarially perturbed to be classified as 'stupa'. Using the DaLLE2 Decoder (Ramesh et al., 2022), we visualize the stages of image embedding purification. The text below each picture annotates the classification results corresponding to each timestep in the purification process.

Additionally, we illustrate the purification path using T-SNE. As depicted in Figure 7, an image sampled from CIFAR-10 originally categorized as 'airplane' was adversarially manipulated to resemble a 'truck,' making it an outlier. Through our CLIPure method, the image is purified towards a high-density area and ultimately reclassified accurately as the 'airplane' category.

## D.2 ANALYSIS FROM TEXTUAL PERSPECTIVE

In this section, we provide additional analysis from a textual perspective as a supplement to the analysis. In Figure 8, we take advantage of the text modality of the CLIP model to understand the semantics of clean and adversarial examples by matching them with closely related words. We

Table 7: Zero-shot performance on 13 datasets against AutoAttack under $\ell_\infty$ threat model with $\epsilon = 2/255$ and $\epsilon = 4/255$. FARE (Schlarmann et al., 2024) and TeCoA (Mao et al., 2022) are trained on $\ell_\infty$ threat model with $\epsilon = 4/255$ for its generally better performance than $\epsilon = 4/255$. Underlined results indicate the best robustness among baselines, while bold text denotes state-of-the-art (SOTA) performance across all methods. The term increase represents the percentage improvement in robustness compared to the best baseline method.

| Eval. | Method | CalTech | Cars | CIFAR10 | CIFAR100 | DTD | EuroSAT | FGVC | Flowers | ImageNet-R | ImageNet-S | PCAM | OxfordPets | STL-10 | Average Zero-shot |
|---|---|---|---|---|---|---|---|---|---|---|---|---|---|---|---|
| clean | CLIP | 83.3 | 77.9 | 95.2 | 71.1 | 55.2 | 62.6 | 31.8 | 79.2 | 87.9 | 59.6 | 52.0 | 93.2 | 99.3 | 73.1 |
| | TeCoA | 78.4 | 37.9 | 79.6 | 50.3 | 38.0 | 22.5 | 11.8 | 38.4 | 74.3 | 54.2 | 50.0 | 76.1 | 93.4 | 54.2 |
| | FARE | 84.7 | 63.8 | 77.7 | 56.5 | 43.8 | 18.3 | 22.0 | 58.1 | 80.2 | 56.7 | 50.0 | 87.1 | 96.0 | 61.1 |
| | **CLIPure-Diff** | 79.9 | 75.5 | 94.9 | 63.8 | 55.2 | 58.2 | 29.3 | 75.0 | 87.5 | 55.9 | 56.8 | 90.4 | 98.4 | 70.8 |
| | **CLIPure-Cos** | 82.9 | 78.6 | 95.6 | 73.0 | 55.4 | 63.3 | 33.2 | 79.3 | 87.7 | 58.3 | 52.0 | 92.5 | 99.6 | 73.2 |
| $\ell_\infty = 2/255$ | CLIP | 0.0 | 0.0 | 0.0 | 0.0 | 0.0 | 0.1 | 0.0 | 0.0 | 0.0 | 0.0 | 0.0 | 0.0 | 0.0 | 0.0 |
| | TeCoA | 69.7 | 17.9 | 59.7 | 33.7 | 26.5 | 8.0 | 5.0 | 24.1 | 59.2 | 43.0 | 48.8 | 68.0 | 86.7 | 42.3 |
| | FARE | 76.7 | 30.0 | 57.3 | 36.5 | 28.3 | 12.8 | 8.2 | 31.6 | 61.6 | 41.6 | 48.4 | 72.4 | 89.6 | 45.9 |
| | **CLIPure-Diff** | 75.1 | 65.9 | 92.5 | 52.6 | 45.9 | 41.5 | 20.8 | 65.8 | 86.5 | 49.8 | 51.4 | 86.2 | 97.9 | 64.0 ↑39.4% |
| | **CLIPure-Cos** | 80.8 | 73.9 | 93.0 | 65.0 | 50.7 | 49.1 | 28.2 | 75.3 | 85.4 | 54.3 | 49.1 | 91.2 | 99.5 | 68.8 ↑45.9% |
| $\ell_\infty = 4/255$ | CLIP | 0.0 | 0.0 | 0.0 | 0.0 | 0.0 | 0.0 | 0.0 | 0.0 | 0.0 | 0.0 | 0.0 | 0.0 | 0.0 | 0.0 |
| | TeCoA | 60.9 | 8.4 | 37.1 | 21.5 | 16.4 | 6.6 | 2.1 | 12.4 | 41.9 | 34.2 | 44.0 | 55.2 | 74.3 | 31.9 |
| | FARE | 64.1 | 12.7 | 34.6 | 20.2 | 17.3 | 11.1 | 2.6 | 12.5 | 40.6 | 30.9 | 48.3 | 50.7 | 74.4 | 32.4 |
| | **CLIPure-Diff** | 74.1 | 66.3 | 90.6 | 51.6 | 45.2 | 42.9 | 18.8 | 65.8 | 83.8 | 48.8 | 52.0 | 85.7 | 97.2 | 63.3 ↑95.4% |
| | **CLIPure-Cos** | 80.1 | 72.2 | 91.1 | 59.1 | 50.1 | 48.4 | 26.1 | 74.8 | 84.6 | 52.4 | 48.9 | 91.0 | 99.4 | 67.4 ↑108.0% |

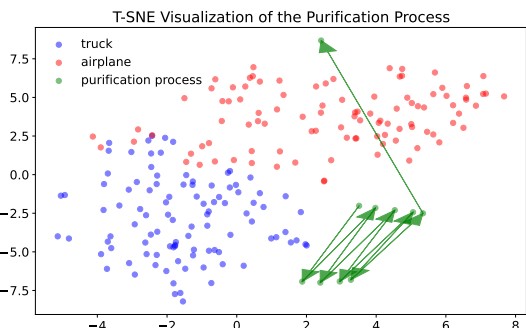

Figure 7: Purification process of the adversarial example attacked from ground truth label 'airplane' to adversarial label 'truck' and purified by our CLIPure-Diff visualized by T-SNE. The image is randomly sampled from the CIFAR-10 dataset.

selected words from ImageNet's categories and supplemented this with an additional 10,000 randomly drawn words from natural language vocabulary for a comprehensive list. We observe that the meanings of the category words ranking high are relatively similar. For clean samples, the distribution of cosine similarity across ranks is relatively stable, whereas the adversarial samples exhibit abnormally high cosine similarity for adversarial categories at top ranks. Figure 9a displays the distribution of cosine similarities between the word embeddings and an image embedding generated from pure noise, across various ranks. We aim to use these similarities to represent the prior probabilities of the words. The results indicate that most samples cluster around a cosine similarity

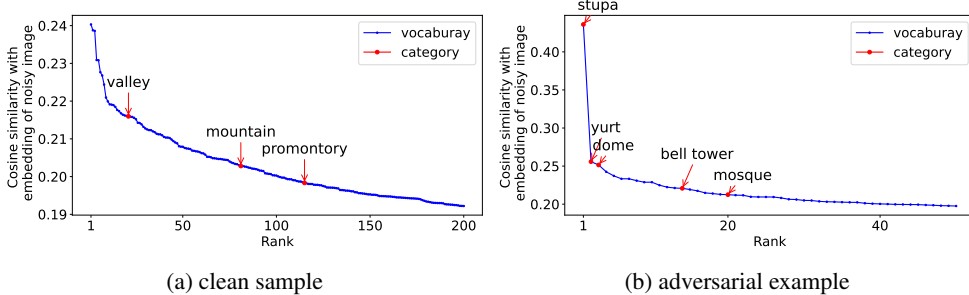

(a) clean sample            (b) adversarial example

Figure 8: Cosine similarity between word and image embeddings of (a) clean and (b) adversarial examples in Figure 6 across different ranks. Blue dots denote 10,000 words sampled from Word2Vec vocabulary (Church, 2017), and red dots denote words from the 1,000 ImageNet categories.

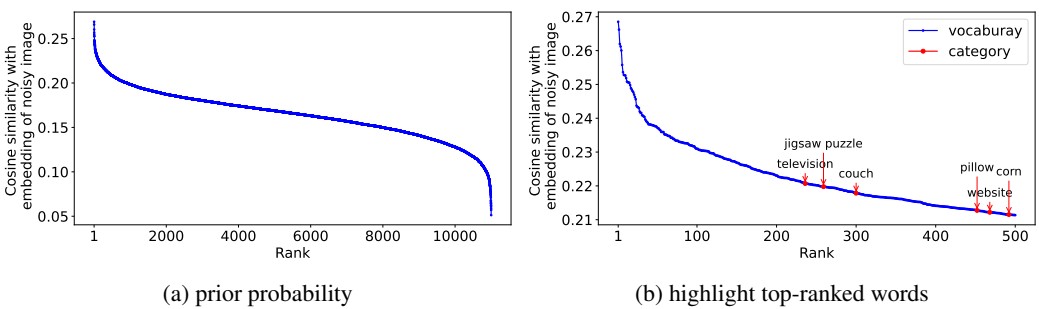

(a) prior probability            (b) highlight top-ranked words

Figure 9: (a)Cosine similarity between words and the image embeddings of a noisy example to illustrate the prior probability of the words. (b) Highlights the top-ranked words within the 1,000 ImageNet categories, emphasizing the most semantically similar words to the noisy image.

of approximately 0.15, with only a few extreme values either much higher or lower, suggesting a normal distribution pattern. This indicates that a majority of the samples have moderate prior probabilities, contrasting with the long-tail distribution often observed in textual data. In Figure 9b, we highlight the top-ranked words in the vocabulary that are closest to the noisy image embedding. Words like "television", which appear frequently in image data, rank high, aligning with our expectations. This abnormal phenomenon in adversarial samples could potentially inspire adversarial detection and more robust adversarial defense methods.

### D.3 COMBINING CLIPURE WITH OTHER STRATEGIES

In this section, we conduct experiments that combine orthogonal strategies, including adversarial training (Schlarmann et al., 2024) and pixel space purification (Chen et al., 2023), as well as incorporating classifier confidence guidance. We carry out evaluations using a sample of 256 images from the ImageNet test set. The experimental setup in this section is aligned with the same configuration as presented in Table 2, detailed in Section 5.1.

**Combination with Adversarial Training (CLIPure+AT).** Orthogonal adversarial training (AT) methods like FARE (Schlarmann et al., 2024) has fintuned CLIP on ImageNet. We combine CLIPure and FARE by purifying image embedding in FARE's latent space and classification via FARE. The experimental results, depicted in Fig. 11a, indicate that the CLIPure-Diff+AT method enhances adversarial robustness in CLIPure-Diff. This improvement could be attributed to FARE providing more accurate likelihood estimates, as it is specifically trained on adversarial examples. When combined with CLIPure-Cos, the outcomes were comparable, likely because the inherent robustness of CLIPure-Cos is already high, leaving limited scope for further enhancement.

**Combination with Pixel Space Purification (CLIPure+LM).** Purification in latent space and pixel space occurs at different stages of processing. Pixel space purification acts directly on the input samples, while latent space purification operates on the latent vectors encoded from the input picture.

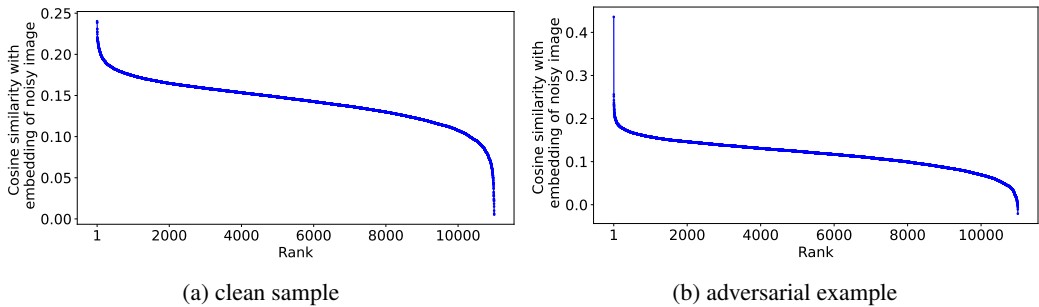

(a) clean sample          (b) adversarial example

Figure 10: Cosine similarity between words and the image embedding of (a) clean sample and (b) adversarial example of the case shown in Figure 6 to illustrate the top-ranked words.

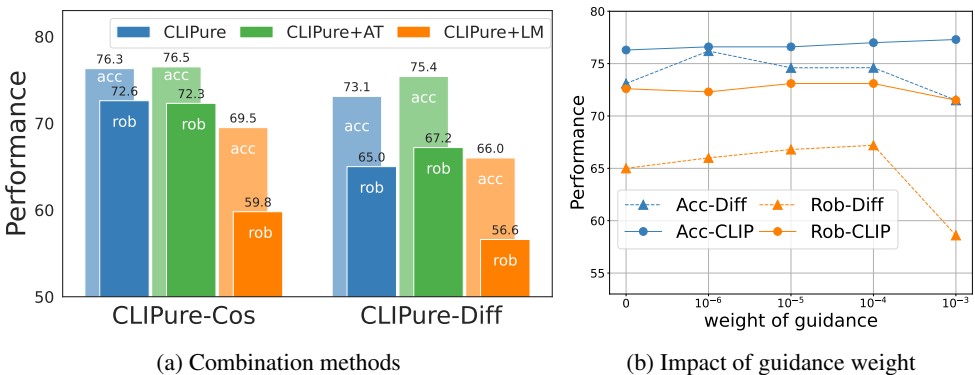

(a) Combination methods          (b) Impact of guidance weight

Figure 11: (a) Accuracy and robustness against AutoAttack with $\ell_\infty = 4/255$ on ImageNet of CLIPure, CLIPure combined with Adversarial Training (AT), and CLIPure combined with pixel space Likelihood Maximization (LM). (b) Performance on ImageNet against AutoAttack with $\ell_\infty = 4/255$ incorporating with classifier confidence guidance.

By combining both methods, we first purify the input samples in pixel space, then pass the purified samples through an encoder to obtain latent vectors, which are further purified in latent space. We mark this combination as CLIPure+LM. For the Likelihood Maximization (LM) method, we chose LM-EDM (Chen et al., 2023) due to its strong performance across pixel space purification methods.

Experimental results, as shown in Fig. 11a, indicate that combining with LM leads to a decrease in both clean accuracy and robustness. This decline may be associated with the challenges of information loss inherent to pixel space purification. Since LM operates directly on the pixels of the image, it is highly sensitive to the degree of purification. However, the adversarial perturbation varies across samples, potentially leading to over-purification thus a decrease in performance. In contrast, purification in latent space with polar coordinates simply involves changing vector directions—altering semantics without reducing semantic information. Thus, it avoids the problem of semantic information loss.

**Combination with Classifier Guidance**. In Eq. 4, we note that the reverse-time SDE associated with the attack method includes not only a likelihood maximization purification term but also a classifier guidance term, represented as $\nabla_{\boldsymbol{x}}\mathcal{L}(\theta; \boldsymbol{x}_t, y_{\text{true}})$. This approach aligns with the classifier confidence guidance proposed by Zhang et al. (2024). We incorporate this classifier guidance term in two versions of CLIPure (CLIPure-Diff and CLIPure-Cos) as described in Algorithm 1. Since the ground truth label $y_{\text{true}}$ is unknown, we employ the predicted label by the CLIP model as a proxy for $y_{\text{true}}$. To address potential inaccuracies in early estimations, classifier guidance was only applied in the final 5 steps of the 10-step purification process.

We evaluate the performance varied with the weight of the guidance term, as depicted in Fig. 11b. It shows that a guidance weight of $10^{-4}$ led to the largest improvement in robustness: CLIPure-

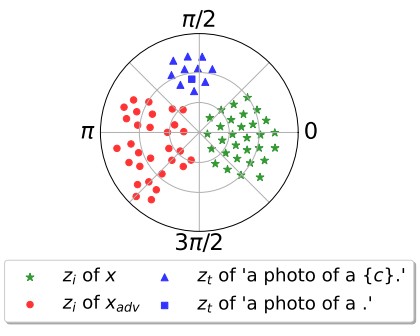

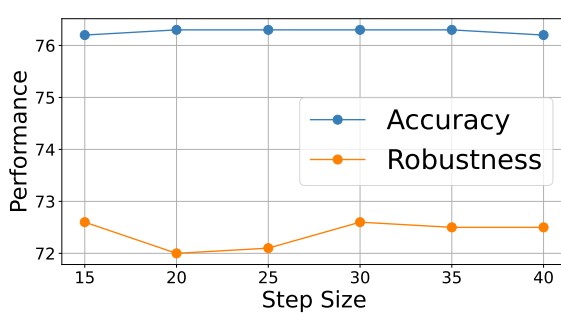

(a) Embeddings in CLIP's latent space          (b) Impact of step size

Figure 12: (a) Visualization of image and text embeddings reduced via PCA and plotted in polar coordinates using t-SNE. $z^i$ and $z^t$ represent the image and text embeddings obtained through CLIP, respectively. Since vector direction signifies semantics, we depict the samples in polar coordinates to emphasize directional properties. (b) Impact of step size during a 10-step purification process using CLIPure-Cos on 1000 ImageNet samples against AutoAttack with $\ell_\infty$ threat model ($\epsilon = 4/255$).

Diff achieved a robustness of 67.2% (an increase of +2.2% over scenarios without guidance), and CLIPure-Cos attained a robustness of 73.1% (an improvement of +0.5% over no guidance).

## D.4    T-SNE VISUALIZATION

In Figure 12a, we display the distribution of image embeddings $z^i$ for both benign samples and adversarial examples from a subset of 30 samples on the CIFAR-10 dataset, after dimensionality reduction using t-SNE. The figure also includes embeddings for 10 category-associated texts (e.g., "a photo of a dog.") $z^t$, as well as the text embedding for a blank template "a photo of a .".

We observe distinct clustering of text vectors, clean image embeddings, and adversarial image embeddings in the space. The blank template, which is used for purification, is positioned at the center of the category text clusters, representing a general textual representation. This arrangement demonstrates that clean and adversarial sample distributions occupy distinct regions in the embedding space, providing a foundational basis for the effectiveness of adversarial sample purification.

## D.5    HYPERPARAMETERS

### D.5.1    IMPACT OF STEP SIZE

We conducted experiments to assess the impact of different step sizes on the effectiveness of purification. Figure 12b shows how the clean accuracy and robustness against AutoAttack ($\ell_\infty = 4/255$) of our CLIPure-Cos method vary with changes in step size. These experiments were carried out on 1,000 samples from the ImageNet test set with 10-step purification. The results indicate that the performance of the method is relatively stable across different step sizes, consistently demonstrating strong effectiveness.

Limited to the computational complexity of CLIPure-Diff, we opt not to conduct a parameter search for this model. Instead, we directly apply the optimal parameters found for CLIPure-Cos.

### D.5.2    IMPACT OF PURIFICATION STEP

As illustrated in Figure 13 (Left), we assess the performance of various purification steps in CLIPure-Cos and CLIPure-Diff. The results indicate that insufficient purification while maintaining acceptable clean accuracy, results in low robustness. As the number of purification steps increases, robustness gradually improves. Continuing to increase the purification steps stabilizes both accuracy and robustness, demonstrating a balance between the two metrics as the process evolves.

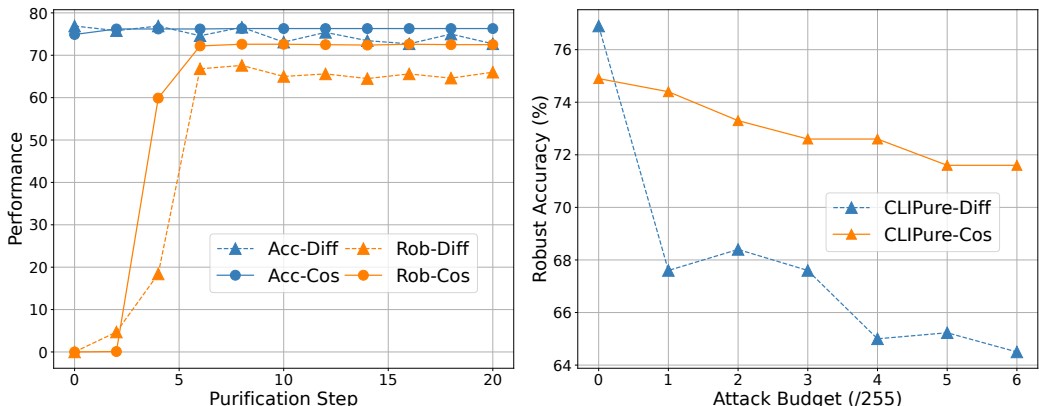

Figure 13: Clean accuracy (marked as "Acc") and robust accuracy (marked as "Rob") across different (Left) purification steps and (Right) attack budgets. "Diff" indicates CLIPure-Diff while "Cos" represents CLIPure-Cos.

## D.6 PERFORMANCE ACROSS ATTACK BUDGET

To further explore the adversarial defense capabilities of CLIPure, we evaluate the robustness of CLIPure-Diff and CLIPure-Cos against various attack budgets, following the settings used in Table 2. As shown in Figure 13 (Right), CLIPure-Diff demonstrates superior clean accuracy (at $\epsilon$=0); however, its robustness decreases as the intensity of the attacks increases. In contrast, CLIPure-Cos exhibits a stronger ability to withstand adversarial perturbations.

