# OpenReview forum: "CLIPure: Purification in Latent Space via CLIP for Adversarially Robust Zero-Shot Classification"
_ICLR.cc/2025/Conference — ICLR 2025 Poster_

### Official Review · Reviewer_xBxY · 2024-10-29

**Soundness:** 3
**Presentation:** 3
**Contribution:** 3
**Rating:** 6
**Confidence:** 3

**Summary:**

The paper propose two variants (i.e., CLIPure-Diff and CLIPure-Cos) for our CLIPure approach, which explore purification in the multi-modal latent space of CLIP. In addition, CLIPure-Cos is the first purification method that is not based on generative models. Experiment results show that purification in multi-modal latent space is promising for zero-shot adversarial robustness.

**Strengths:**

1. Conducting purification in multi-modal latent space for adversarially robust zero-shot classification is novel.
2. The overall organization is reasonable, and the writing is good.
3. Sufficient experiments are performed, and the proposed method exceeds the comparison methods.

**Weaknesses:**

1. Line 184: $p_{ben}(X)$ is not mentioned in Eq. (3).
2. I wonder why $KL((p(x_{adv})||(p(x_{ben}))$ in Eq. (6) can excel at detecting out-of-distribution adversarial examples. Please further explain it.
3. In the experiment setting, authors should provide more training and testing details, and  Hyperparameter settings.
4. Providing more visual results about the Purification process like Figure 5 would be better.

**Questions:**

Please refer to the Weaknesses.

---

> ### Author Response · Authors · 2024-11-22
>
> **We appreciate the time and effort of the reviewer.** In response to the issues raised in the review, we offer the following replies:
>
> **Q1**: Line 184: $p_{\text{ben}}(x)$ is not mentioned in Eq. (3)
>
> **A1**: In Line 179, we specified that $p(x)$ represents $p_{\text{ben}}(x)$ for simplicity, so $p_{\text{ben}}(x)$ is referred to as $p(x)$ in Eq.(3). We have corrected this notation in Line 184 in the revised version. Thank you for bringing this to our attention.
>
> **Q2**: I wonder why $\text{KL}((p(x_{\text{adv}}) || (p(x_{\text{ben}}))$ in Eq. (6) can excel at detecting out-of-distribution adversarial examples. Please further explain it.
>
> **A2**: Firstly, $p(x)$ models the likelihood of sample x, representing the probability of generating image x. For models trained on clean samples, the likelihood (probability) of generating clean samples is higher than that for out-of-distribution samples, such as adversarial examples. This difference leads to a discrepancy between $p(x_{\text{adv}})$ for generating adversarial samples $x_{\text{adv}}$ and $p(x_{\text{ben}})$ for generating benign samples $x_{\text{ben}}$. The KL divergence, $\text{KL}(p(x_{\text{adv}}) \| p(x_{\text{ben}}))$, measures the distributional difference between clean and adversarial samples. When considering a trained generative model estimating likelihood $p(x)$, higher KL divergence indicates a stronger ability of the model to discriminate between adversarial and clean samples by effectively modeling the boundary between clean and adversarial examples better, thereby potentially achieving more effective purification.
>
>
> **Q3**: In the experiment setting, authors should provide more training and testing details and Hyperparameter settings.
>
> **A3**: Considering the space limit, we place detailed descriptions of our experimental settings in Appendix C in the original version, titled "More Experimental Settings". Thank you for your suggestion; we additionally include the experimental setting for baselines in Appendix C.3 in the revised version.
>
> **Q4**: Providing more visual results about the purification process like Figure 5 would be better.
>
> **A4**: We appreciate your valuable feedback. We include the visualizations of the purification trajectories by T-SNE in Figure 8 to illustrate the purification path from the adversarial category back to the correct category. We hope these additions provide a clearer visual understanding of the purification process to demonstrate the effectiveness of CLIPure.
>
>
> **We appreciate your efforts and are open to further discussion if you have any additional concerns.**

---

> > ### Comment · Reviewer_xBxY · 2024-11-25
> > **Official Comment by Reviewer xBxY**
> >
> > Thanks for the authors' response. The rebuttal has addressed most of my concerns. I keep my positive score.

---

> > > ### Author Response · Authors · 2024-11-25
> > >
> > > We sincerely thank you for your positive feedback. We remain open and responsive to any further discussions until the end of the discussion stage.

---

### Official Review · Reviewer_3KPg · 2024-10-29

**Soundness:** 3
**Presentation:** 3
**Contribution:** 2
**Rating:** 6
**Confidence:** 1

**Summary:**

This paper presents CLIPure, an approach for building adversarially robust zero-shot image classifiers based on CLIP. CLIPure performs purification in the latent space of CLIP to enhance robustness against adversarial attacks. It introduces two purification methods: CLIPure-Diff, which uses DALLE 2's diffusion prior to pure latent vectors ,and CLIPure-Cos, which performs purification by computing cosine similarity between the embeddings of an image and textual embedding of the blank templates. This paper conducts extensive experiments on CIFAR-10, ImageNet, and other 13 datasets and the proposed method demonstrates strong performance, exceeding previous SOTA results.

**Strengths:**

1. The proposed method performs purification in CLIP's latent space, which is technically sound.

2. Based on the cosine similarity used in CLIP, this paper further proposes a purification method that is not based on generative models, which improves the computational efficiency.

**Weaknesses:**

1. The motivation is not strong enough. It seems that the key motivation is simply that performing purification in the latent space of CLIP is promising to achieve improved performance. However, there are only a few discussions about why it is necessary to perform purification methods for CLIP in its latent space and what issues exist with previous methods for CLIP.

2. Some baseline methods need further clarification. It is unclear whether the authors pre-trained or trained the baselines themselves or used publicly available versions. If off-the-shelf models are used, it is necessary to specify the references and discuss whether the difference in training data influences the comparisons. If the authors conducted the training themselves, which dataset is used and what is the setting for the training?

3. It seems that the experimental settings for baselines are not clear enough. For the compared purification methods, the base generative models should be unified for a fair comparison. Especially, it would be better to compare to the purification method with DiffPure based on DALLE2. For the AT methods, as the proposed method is specifically designed for CLIP, further discussion is needed on the comparability between different base models.

**Questions:**

I am not familiar with this field. Please refer to the weaknesses for my major concerns.

---

> ### Author Response · Authors · 2024-11-23
> **Reply (1/2)**
>
> **Reply (1/2)**
>
> **We appreciate the time and effort of the reviewer.** In response to the issues raised in the review, we offer the following replies:
>
> **Q1**: The motivation is not strong enough. It seems that the key motivation is simply that performing purification in the latent space of CLIP is promising to achieve improved performance. However, there are only a few discussions about why it is necessary to perform purification methods for CLIP in its latent space and what issues exist with previous methods for CLIP.
>
> **A1**: Our motivation is to create an **adversarially robust zero-shot image classifier that can efficiently and accurately classify unseen  examples meanwhile defending against any unforeseen adversarial attacks, addressing significant challenges in real-world safety-critical scenarios.** To achieve this, we focus on two main issues: zero-shot classification and defense against unseen attacks.
>
> Regarding **zero-shot classification**, we start with CLIP, a popular, efficient, and effective zero-shot classifier, that has decent zero-shot classification performance. Despite its efficacy on clean samples, it is vulnerable to adversarial attacks. Existing defenses like TeCoA [1] and FARE [2], which involve adversarial training of CLIP—compared in Tables 2 and 4—often fail to generalize to new, unseen attacks and can degrade the intrinsic zero-shot capabilities of CLIP due to fine-tuning on specific datasets.
>
> To effectively **defend against any unforeseen attacks**, we employ adversarial purification instead of adversarial training. To explore which space would be better for purification, we derive purification risk through bidirectional Stochastic Differential Equations (SDEs). The theoretical results inspire us to purify adversarial examples in CLIP's latent space (detailed in Sections 4.1 and 4.2).
>
> Based on these insights, we developed CLIPure, a novel method designed to achieve adversarially robust zero-shot classification efficiently against unseen attacks without additional training.
>
>
> **Q2**: Some baseline methods need further clarification. It is unclear whether the authors pre-trained or trained the baselines themselves or used publicly available versions. If off-the-shelf models are used, it is necessary to specify the references and discuss whether the difference in training data influences the comparisons. If the authors conducted the training themselves, which dataset is used and what is the setting for the training?
>
> **A2**: We appreciate your valuable feedback. To address the clarity regarding baseline methods, we have included a detailed description of the settings for the baselines in **Appendix C.3 of our revised paper**. For experiments on CIFAR-10 (detailed in Table 1) and ImageNet (detailed in Table 2), we utilized the results from each baseline's original paper, due to the same attack evaluation settings as ours to ensure consistency. For the experiments involving multiple datasets shown in Table 4 in the original version, we used the checkpoints provided by TeCoA [1] and FARE [2] and evaluated them under our threat model, consistent with CLIPure.
> Additionally, for variants of CLIPure like LM-StableDiffusion and LM-DaLLE2.Decoder, we employed off-the-shelf models without additional training and applied them to the purification method. This allows us to make a fair comparison.
>
>
> **We appreciate your efforts and are open to further discussion if you have any additional concerns.**

---

> ### Author Response · Authors · 2024-11-23
> **Reply (2/2)**
>
> **Reply (2/2)**
>
> **We appreciate the time and effort of the reviewer.** In response to the issues raised in the review, we offer the following replies:
>
> **Q3**: It seems that the experimental settings for baselines are not clear enough. For the compared purification methods, the base generative models should be unified for a fair comparison. Especially, it would be better to compare to the purification method with DiffPure based on DALLE2. For the AT methods, as the proposed method is specifically designed for CLIP, further discussion is needed on the comparability between different base models.
>
> **A3**: We experiment on various generative-model-based baselines to compare purification in different modeling spaces, such as pixel space, uni-modal space, and multi-modal space, as stated in Section 4.2.  Following your valuable advice, we have incorporated the results of the DiffPure-DaLLE2.Decoder into our experiments, in line with the setup used in Table 2. It shows that the Likelihood Maximization (LM) purification method in pixel space performs comparably to the DiffPure method when using DaLLE2.Decoder as the backbone.
>
> | Method              | Clean Accuracy  | Robust Accuracy  |
> |--|:--:|:--:|
> | LM-DaLLE2.Decoder   | 36.9 | 9.2  |
> | DiffPure-DaLLE2.Decoder | 31.2 | 9.0  |
> | CLIPure-Diff        | 73.1 | 65.0 |
> | CLIPure-Cos         | 76.3 | **72.6** |
>
> If we correctly understand your concerns regarding AT baselines, we have included **AT baselines based on the CLIP model as well as other classifiers such as ConvNeXt-L, Swin-L, and WideResNet-70-16.** Our evaluation method maintains consistency across different baselines for a fair comparison.
>
> Regarding the base models for AT methods, we included TeCoA and FARE, which specifically involve adversarially-trained CLIP under scenarios of known attacks. Additionally, our baselines encompass state-of-the-art AT methods tailored to each dataset, such as AT-DDPM [3] and AT-EDM [4] based on WideResNet-70-16 on CIFAR-10 dataset, listed in Table 1, and AT-ConvNeXt-L [5] and AT-Swin-L [6] on ImageNet in Table 2. These methods are trained adversarially on specific datasets against known attack methods. In contrast, our CLIPure approach is evaluated without additional training on specific attacks used in testing. This difference underscores the novelty and applicability of CLIPure, making it robust against unforeseen attacks, which is a more realistic scenario.
>
>
> **We appreciate your efforts and are open to further discussion if you have any additional concerns.**
>
> [1] Understanding zero-shot adversarial robustness for large-scale models, ICLR 2023
>
> [2] Robust CLIP: Unsupervised Adversarial Fine-Tuning of Vision Embeddings for Robust Large Vision-Language Models, ICML 2024
>
> [3] Fixing data augmentation to improve adversarial robustness, NeurIPS 2021
>
> [4] Better diffusion models further improve adversarial training, ICML 2023
>
> [5] Revisiting adversarial training for imagenet: Architectures, training and generalization across threat models, NeruIPS 2024
>
> [6] A comprehensive study on robustness of image classification models: Benchmarking and rethinking, IJCV 2024

---

### Official Review · Reviewer_Hwgg · 2024-11-01

**Soundness:** 3
**Presentation:** 2
**Contribution:** 3
**Rating:** 6
**Confidence:** 4

**Summary:**

In this work,the author propose CLIPure, which conducts purification in CLIP's latent space for adversarially robust zero-shot classification.CLIPure leverages the image encoder and text encoder of the CLIP model to achieve effective purification in the latent space.By minimizing the KL divergence between the purification process and the adversarial process, CLIPure reduces purification risks and enhances the model's robustness against adversarial attacks. The algorithm includes two versions: CLIPure-Diff and CLIPure-Cos, which are based on the DiffusionPrior model and the cosine similarity metric of CLIP respectively. Experimental results demonstrate that CLIPure significantly outperforms existing methods when defending against AutoAttack on multiple datasets.

**Strengths:**

This paper introduces CLIPure-Diff and CLIPure-Cos, which perform purification in CLIP's latent space rather than in pixel space. Both methods exhibit significantly larger KL divergence between adversarial and benign sample distributions when compared to methods operating in pixel space and uni-modal latent space. Furthermore, CLIPure-Cos enhances defense efficiency by not relying on generative models. Results demonstrate that CLIPure significantly improves the SOTA robustness.

**Weaknesses:**

1、Although the article conducts numerous experiments, the selected comparison methods are not entirely appropriate. For instance, the adversarial training method in Table 2 is intended for defending against attacks on the text functionality of CLIP, rather than for zero-shot defense based on CLIP.
2、It is recommended to conduct further comparisons with models in the same direction, such as "Understanding zero-shot adversarial robustness for large-scale models" (ICLR 2023) and "Pre-trained Model Guided Fine-Tuning for Zero-Shot Adversarial Robustness" (CVPR 2024).

**Questions:**

1.The paper mentions using 80 different description templates to enhance stability. How were these templates selected? Is it possible to optimize the selection of these templates through an automated approach?


2. Regarding the defensive effectiveness of the model, how does it perform under different threat levels (ϵ values)? Is there a detailed analysis available?

3、The article mentions that the selected purification step is 10 steps, but it lacks relevant experimental evidence to support this. How is this number determined? Is there experimental proof that this is the optimal number of steps, or is there a potentially better number of steps?

---

> ### Author Response · Authors · 2024-11-22
> **(Reply 1/2)**
>
> **(Reply 1/2)**
>
> **We appreciate the time and effort of the reviewer.** In response to the issues raised in the review, we offer the following replies:
> **Q1**: Although the article conducts numerous experiments, the selected comparison methods are not entirely appropriate. For instance, the adversarial training method in Table 2 is intended for defending against attacks on the text functionality of CLIP, rather than for zero-shot defense based on CLIP.
>
> **A1**: **Zero-shot classification is the primary downstream task targeted by both FARE [1] and TeCoA [2]** (the adversarial training method in Table 2). **As stated in FARE**, "_We propose an unsupervised adversarial fine-tuning scheme to obtain a robust CLIP vision encoder, which yields robustness on all vision downstream tasks (LVLMs, **zero-shot classification**) that rely on CLIP._" Therefore, FARE is explicitly designed to enhance defense capabilities not only against attacks targeting text functionality but also for zero-shot classification. **TeCoA** ("Understanding zero-shot adversarial robustness for large-scale models", ICLR 2023) **is the work you highlighted in Q2 as needing comparison**. These baselines are thus directly comparable and relevant to evaluating the effectiveness of our approach.
>
> **Q2**: It is recommended to conduct further comparisons with models in the same direction, such as "Understanding zero-shot adversarial robustness for large-scale models" (ICLR 2023) and "Pre-trained Model Guided Fine-Tuning for Zero-Shot Adversarial Robustness" (CVPR 2024).
>
> **A2**: **TeCoA** (Understanding zero-shot adversarial robustness for large-scale models, ICLR 2023) **is included in our experiments in Table 2 and Table 5** (Table 4 in the original version). According to your valuable feedback, we further incorporate a comparison with the second paper you mentioned, "Pre-trained Model Guided Fine-Tuning for Zero-Shot Adversarial Robustness" (CVPR 2024) [3].
>
> We follow their experimental setup, using the ViT-B-32 version of the CLIP model and defending against the $\ell_{\infty}$ threat model with $\epsilon=1/255$ on 9 datasets.
> Our results shown in the following table indicate that:
> - CLIPure, without additional training, **preserves the zero-shot classification accuracy of CLIP**.
> - CLIPure **outperforms PMG on all 9 datasets**.
>
> "Rob" and "Acc" for robustness and accuracy respectively:
>
> |                    | Cars   | CIFAR10 | CIFAR100 | DTD    | EuroSAT | FGVC  | Flowers | OxfordPets | STL-10 | Average |
> |:---:|--------|:---:|:---:|:---:|:---:|:---:|:---:|:---:|:---:|:---:|
> | **Rob**            |        |         |          |        |         |       |         |            |        |         |
> | CLIP               | 2.34   | 42.18   | 17.57    | 18.75  | 3.84    | 0.39  | 4.88    | 8.98       | 67.77  | 18.52   |
> | PMG-AFT            | 14.64  | 66.99   | 38.28    | 34.38  | 24.15   | 3.91  | 23.43   | 33.59      | 76.17  | 35.06   |
> | VPT-PMG-AFT        | 5.46   | 50.19   | 22.07    | 3.45   | 11.08   | 2.73  | 16.85   | 18.55      | 64.45  | 21.65   |
> | **CLIPure-Cos**        | **57.60**  | **88.70**   | **56.10**    | **43.00**  | **44.30**   | **19.10** | **65.20**   | **85.20**      | **97.00**  | **61.80**   |
> | **Acc**            |        |         |          |        |         |       |         |            |        |         |
> | CLIP               | 55.27  | 89.52   | 62.50    | 41.79  | 50.84   | 8.70  | 48.82   | 85.93      | 97.46  | 60.09   |
> | PMG-AFT            | 48.24  | 83.98   | 58.39    | 41.02  | 35.28   | 6.25  | 42.96   | 84.76      | 92.97  | 54.87   |
> | VPT-PMG-AFT        | 16.99  | 67.38   | 33.78    | 13.92  | 13.92   | 7.81  | 17.18   | 49.60      | 81.64  | 33.58   |
> | **CLIPure-Cos**        | 61.00  | 90.60   | 60.70    | 46.88  | 49.70   | 21.00 | 67.20   | 87.20      | 97.40  | 64.63   |
>
> Thank you for bringing this to our attention. We have included it in our related work section.
>
> **Q3**: The paper mentions using 80 different description templates to enhance stability. How were these templates selected? Is it possible to optimize the selection of these templates through an automated approach?
>
> **A3**: We use the 80 templates provided in the **OpenAI CLIP repository**. While there may be more optimal strategies for selecting templates, this is not the primary focus of our paper. However, you raise an interesting point, and we will consider to explore further the impact of template selection on robustness in future work.
>
>
> **We appreciate your efforts and are open to further discussion if you have any additional concerns.**

---

> ### Author Response · Authors · 2024-11-22
> **(Reply 2/2)**
>
> **We appreciate the time and effort of the reviewer.** In response to the issues raised in the review, we offer the following replies:
>
> **Q4**: Regarding the defensive effectiveness of the model, how does it perform under different threat levels ($\epsilon$ values)? Is there a detailed analysis available?
>
> **A4**: According to your valuable advice, we evaluate the robustness of CLIPure-Diff and CLIPure-Cos against varying adversarial attack budgets $\epsilon$ on the ImageNet dataset, following the setting used in Table 2. We also include a line graph in **Figure 13 (Right)** of our revised paper to visually depict the result. It reveals that CLIPure-Diff, while demonstrating superior clean accuracy at $\epsilon=0$, experiences a significant decrease in robustness by 12.4% as the attack intensity increases to $\epsilon=6/255$. Another version, CLIPure-Cos, shows a more stable performance with a smaller decrease in robustness, dropping by 3.3% under the same increase in threat levels from $\epsilon=0$ to $\epsilon=6/255$.
>
> |   $\epsilon $ (/255)            | 0    | 1    | 2    | 3    | 4    | 5    | 6    |
> |---------------|:---:|:---:|:---:|:---:|:---:|:---:|:---:|
> | CLIPure-Cos   | 74.9 | 74.4 | 73.3 | 72.6 | 71.6 | 71.6 | 71.6 |
> | CLIPure-Diff  | 76.9 | 67.6 | 68.4 | 67.6 | 65.0 | 65.2 | 64.5 |
>
> **Q5**: The article mentions that the selected purification step is 10 steps, but it lacks relevant experimental evidence to support this. How is this number determined? Is there experimental proof that this is the optimal number of steps, or is there a potentially better number of steps?
>
> **A5**: We initially tested the robustness of different purification steps on a smaller dataset with 50 samples, leading us to select 10 steps. Thank you for your suggestion. In further evaluations, we evaluate the robustness of CLIPure-Cos and CLIPure-Diff across various purification steps. Our findings indicate that CLIPure-Cos achieves optimal results with 10 steps, while CLIPure-Diff shows improved robustness at 8 steps. We plan to update our results using CLIPure-Diff with an 8-step purification process. Additionally, we depict the line graph in **Figure 13 (Left)** of our revised paper to visually represent these results.
>
> | Purification Step | 0    | 2    | 4    | 6    | 8    | 10   | 12   | 14   | 16   | 18   |
> |---------------|:---:|:---:|:---:|:---:|:---:|:---:|:---:|:---:|:---:|:---:|
> | **CLIPure-Cos Acc** | 74.9 | 76.2 | 76.2 | 76.2 | 76.3 | 76.3 | 76.3 | 76.3 | 76.3 | 76.3 |
> | **$\qquad \qquad \quad \ $ Rob** | 0.0  | 0.1  | 59.9 | 72.2 | **72.6** | **72.6** | 72.5 | 72.4 | 72.6 | 72.5 |
> | **CLIPure-Diff Acc**| 76.9 | 75.8 | 76.9 | 74.6 | 76.6 | 73.1 | 75.4 | 73.44| 72.7 | 75.0 |
> | **$\qquad \qquad \quad \ $ Rob**| 0.0  | 4.7  | 18.4 | 66.8 | **67.6** | 65.0 | 65.6 | 64.5 | 65.6 | 64.6 |
>
>
> **We appreciate your efforts and are open to further discussion if you have any additional concerns.**
>
>
> [1] Robust CLIP: Unsupervised Adversarial Fine-Tuning of Vision Embeddings for Robust Large Vision-Language Models, ICML 24
>
> [2] Understanding zero-shot adversarial robustness for large-scale models, ICLR 23
>
> [3] Pre-trained Model Guided Fine-Tuning for Zero-Shot Adversarial Robustness, CVPR 2024

---

> > ### Comment · Reviewer_Hwgg · 2024-11-27
> >
> > Thank you for the author's reply. In your supplementary experiments,  CLIPure has demonstrated its performance in defending against purification risks. Your rebuttal has basically resolved my doubts, and I have raised my score.

---

> > > ### Author Response · Authors · 2024-11-27
> > >
> > > We sincerely thank you for your positive feedback. We remain open and responsive to any further discussions until the end of the discussion stage.

---

### Official Review · Reviewer_7kUG · 2024-11-02

**Soundness:** 2
**Presentation:** 2
**Contribution:** 2
**Rating:** 6
**Confidence:** 5

**Summary:**

This paper enhances adversarial robustness by purifying images in latent space. The experiments show a substantially improvement.

**Strengths:**

1. Good performance.

2. Clear figures with bright color.

**Weaknesses:**

Although the method shows a significant improvement, it still has many non-ignorable problems:

1. The motivation and contribution are too verbose, as well as the background description.
The authors should compact it.
For example, The prompt content of _"... by matching an image with text prompts “a photo of <class-name>”. "_ in abstract can be completely removed.

2. The method seems just do purify in feature space instead of in pixel space.

3. Lack a figure to illustrate the process of the method.

4. The theoretically analysize about purification risk is difficult to understand. It seems just a SDE?

5. And the analysize only is from CLIP, how can you extend it to diffusion model?

6. The definition of Zero-shot Learning is not correct.
In the original paper of CLIP, it has clearly claimed they changed the definition of ZSL from class-level to dataset-level.

_"In computer vision, zero-shot learning usually refers to the study of generalizing to unseen object categories in image classification (Lampert et al., 2009). We instead use the term in a broader sense and study generalization to unseen datasets."_

7. Where is the ablation study?

8. From table 3, we can clearly see CLIPure-Diff also increases the inference time significantly. And the efficiency comparsion misses the result of  FARE.

**Questions:**

See weakness

**Details Of Ethics Concerns:**

The paper involves a pretrained Stable Diffusion. Thus, the bias problem must be considered.

---

> ### Author Response · Authors · 2024-11-23
> **Reply (1/2)**
>
> **Reply (1/2)**
>
> **We appreciate the time and effort of the reviewer.** In response to the issues raised in the review, we offer the following replies:
>
> **Q1**: The motivation and contribution are too verbose, as well as the background description. The authors should compact it.
>
> **A1**: We intended to ensure that readers from diverse backgrounds could thoroughly understand our work. We appreciate your feedback and will consider revising the background description.
>
> **Q2**: The method seems just do purify in feature space instead of in pixel space.
>
> **A2**: As far as we know, existing adversarial purification methods all purify adversarial examples in pixel space. **The efficacy of latent space purification, particularly in multi-modal latent space, remains unexplored and unknown**. Our work is **the first to propose purification in multi-modal latent space** and is grounded on CLIP for robust zero-shot classification. Moreover, **our CLIPure-Cos**, which innovatively estimates likelihood using cosine similarity to null templates, is **the first purification method not based on generative models**, substantially improving defense efficiency besides state-of-the-art robustness.
>
> **Q3**: Lack a figure to illustrate the process of the method.
>
> **A3**: Due to space constraints, we initially described the CLIPure purification process in Algorithm 1 within the main paper, rather than including a figure. According to your advice, we **add Figure 5 to our revised paper to illustrate the CLIPure process** and hope this addition will help readers better understand our methodology.
>
>
> **Q4**: The theoretical analysis of purification risk is difficult to understand. It seems just an SDE?
>
> **A4**:  **We use Stochastic Differential Equations (SDEs) to formulate the attack and purification process to derive the purification risk, which inspires us to explore more effective purification strategies within the latent space of the multi-modal model.**
>
> Specifically, we employ bidirectional SDEs to model the adversarial attack process (i.e., adding perturbations to benign examples) through a forward SDE, and the purification process (i.e., denoising adversarial examples) through a reverse SDE. This dual-SDE framework allows us to accurately quantify purification risk by comparing the difference between the attack trajectory and the purification path as shown in Eq. (5), since effective purification is essentially the inverse operation of the attack process.
>
> We further derive the lower bound of the purification risk in Eq. (6), which is influenced by two factors: 1) the smoothness of the log-likelihood function at adversarial examples, possibly extended by the sample dimension, as indicated by the $\ell_2$ norm of $\nabla \log p(\boldsymbol{x}_{\text{adv}})$, and 2) the differences between the likelihoods of clean and adversarial samples in the benign example space.
>
> These insights motivate us to purify adversarial examples in CLIP's latent space since multi-modal latent representations are denser and smoother compared to pixel space and own better discriminative representation than uni-modal models.
>
>
> **Q5**: And the analysis is only from CLIP, how can you extend it to the diffusion model?
>
> **A5**: Our analysis of purification risk is general and can be applied to any model's attack and defense processes, not limited to CLIP or diffusion models. This is because our modeling focuses on the adversarial attack and defense mechanisms without making specific assumptions about the model itself. We hope this addresses your concern. If you have further questions or need more clarification, please feel free to continue the discussion.
>
> **Q6**: The definition of Zero-shot Learning is not correct. In the original paper of CLIP, it has clearly claimed they changed the definition of ZSL from class-level to dataset-level.
> "_In computer vision, zero-shot learning usually refers to the study of generalizing to unseen object categories in image classification (Lampert et al., 2009). We instead use the term in a broader sense and study generalization to unseen datasets._"
>
> **A6**: **Our paper aligns with the definition of Zero-Shot Learning (ZSL) as used by CLIP, which can generalize to not only unseen categories but also unseen datasets.**
> As you highlighted, the generalization at the dataset level is broader than at the class level, as it encompasses not only new categories but also variations in samples among datasets—similar to understandings of zero-shot learning (ZSL) in other fields such as natural language processing (NLP).  Moreover, CLIPure can adapt to unseen attacks as well as defend against unseen adversarial attacks, thereby maintaining its effectiveness across various new and challenging scenarios. This capability enhances the practical applicability and effectiveness of CLIPure in real-world settings.
>
>
> **We appreciate your efforts and are open to further discussion if you have any additional concerns.**

---

> ### Author Response · Authors · 2024-11-23
> **Reply (2/2)**
>
> **Reply (2/2)**
>
> **We appreciate the time and effort of the reviewer.** In response to the issues raised in the review, we offer the following replies:
>
> **Q7**: Where is the ablation study?
>
> **A7**: Our ablation studies include:
>
> - **CLIP without purification**: We evaluate the baseline performance of the CLIP model without any purification mechanism. The results are detailed in Table 1, Table 2, and Table 4 to highlight the impact and necessity of purification.
>
> - **CLIPure without embedding normalization**: We examine the effects of removing embedding normalization from CLIPure, with findings discussed in Section 4.3. This helps to evaluate the necessity of this component as CLIP  aligns image and text embeddings by their cosine similarities ignoring their vector lengths.
>
> - **Additional variants for comparison**:
>   - **Purification in pixel space**: We compare CLIPure with methods including DiffPure [2], LM-EDM [3], and LM-DaLLE2.Decoder, which performs purification in pixel space. The comparative results are displayed in Table 1 and Table 2.
>   - **Purification in uni-modal model's latent space**: We also explore purification using a uni-modal model's latent space, specifically with LM-StableDiffusion, and presented these results in Table 1.
>
> If you have other suggestions for further ablation studies or additional comparisons that could enhance our analysis, please let us know, and we will try our best to incorporate them into our paper.
>
> **Q8**: From table 3, we can clearly see CLIPure-Diff also increases the inference time significantly. And the efficiency comparison misses the result of FARE.
>
> **A8**: Though CLIPure-Diff (inferior to the other variant CLIPure-Cos) increases inference time compared to CLIP, it still significantly outperforms SOTA baselines and achieves much higher efficiency than traditional purification methods in pixel space such as DiffPure and LM-EDM, i.e., approximately 221 times and 24 times, respectively. This is because purification in latent space operates on image embedding, which is of low dimensionality than pixel images.
>
> We did not directly compare the inference efficiency of TeCoA [4] and FARE [5] because their computational overhead primarily lies in training rather than inference. Their inference time is consistent with that of the original CLIP model.
>
>
> **Ethics concerns**: The paper involves a pretrained Stable Diffusion. Thus, the bias problem must be considered.
>
> **Clarification**: We employ the pretrained Stable Diffusion (SD) model **not for image generation but solely for comparative analysis within its latent space**. Additionally, when using the diffusion model for purification, the alterations made to images are minimal and imperceptible to humans; we do not generate new images. Therefore, we believe that our usage does not involve ethical concerns related to image generation biases or the creation of misleading or harmful content.
>
>
> **We appreciate your efforts and are open to further discussion if you have any additional concerns.**
>
> [1] What makes multi-modal learning better than single (provably), NeurIPS 2021
>
> [2] Diffusion Models for Adversarial Purification, ICML 2022
>
> [3] Robust Classification via a Single Diffusion Model, ICML 2024
>
> [4] Understanding zero-shot adversarial robustness for large-scale models, ICLR 2023
>
> [5] Robust CLIP: Unsupervised Adversarial Fine-Tuning of Vision Embeddings for Robust Large Vision-Language Models, ICML 2024

---

> > ### Comment · Reviewer_7kUG · 2024-11-24
> > **Official Comment from Reviewer 7kUG**
> >
> > I appreciate the response from authors, but my concerns are still existing.
> >
> > 1. Please provide the results or related literature that can support your claim "CLIP can generalize to unseen categories".
> >
> > 2. Please further enhance your contribution. Although you are the first to propose purification in multi-modal latent space, but  purification/disturbance in uni-modal latent space has existing for a long time.
> > If your work just moves purification from uni-model space to multi-model space or from multi-model pixel space to multi-model latent space, I suggest the contribution is very incremental.
> >
> > 3. Please provide the revised version addressing my Q1 before the discussion close. I think the current version still is hard to read.

---

> ### Author Response · Authors · 2024-11-25
>
> **We appreciate your time and further feedback.** In response, we offer the following replies:
>
> **Q1**: Please provide the results or related literature that can support your claim "CLIP can generalize to unseen categories".
>
> **A1**: We regret that the statement "CLIP can generalize to unseen categories" in our last response is inaccurate. We intended to say that CLIP can perform zero-shot classification on various categories and data. CLIP can classify an image by measuring its similarity with templates such as "a photo of {category}" **without being trained with any image classification labels, as long as the category text (e.g., "dog") appears in the pre-trained corpus.** From this perspective, it is considered as a zero-shot classifier. For images whose category text description has never occurred, it cannot perform classification on such data. Since CLIP has been trained on a tremendous amount of data, such cases should be rare and it can perform classification on a wide range of categories decently, as shown in our experiments on diverse datasets such as ImageNet, Flowers, CIFAR-100, OxfordPets, Cars, etc.
>
> **Q2**: Please further enhance your contribution. Although you are the first to propose purification in multi-modal latent space, but purification/disturbance in uni-modal latent space has existing for a long time. If your work just moves purification from uni-model space to multi-model space or from multi-model pixel space to multi-model latent space, I suggest the contribution is very incremental.
>
> **A2**: **To the best of our knowledge, this is the first work that proposes adversarial purification in latent space, including both unimodal and multimodal latent space.** According to our theoretical derivation of purification risk, we presume that multimodal latent space should be more promising than unimodal, so we primarily explore CLIP's latent space and emphasize our contribution to purification in multimodal latent space.
>
> We summarize our **contributions** as:
> - We **theoretically formulate purification risk** and show that the lower bound of purification risk is influenced by a) the smoothness of the log-likelihood function at adversarial examples and possibly the sample dimension, and b) the differences between the likelihood of clean and adversarial samples in the benign example space.
> - As far as we know, we are **the first to propose and explore adversarial purification in latent space**. We also compare unimodal latent purification, multimodal latent purification, and multimodal pixel purification under the same experimental settings to show the superior quality of multi-modal latent space in terms of purification risk.
> - Based on CLIP, we propose **the first purification model (i.e., CLIPure-Cos) not using cost-intensive generative models to estimate sample likelihood**. We model sample likelihood by its cosine similarity with anchor templates and propose to normalize image embeddings to unit vectors during likelihood maximization, which diminishes the effect of vector length and is critical for effective purification.
> - Our CLIPure-Cos achieves **remarkable zero-shot adversarial robustness (with very small gaps compared to clean accuracy) against unseen attacks on a wide range of datasets**, e.g., boosting SOTA robustness from 59.6% to 72.6% on ImageNet, 71.7% to 91.1% on CIFAR10, and producing 108% average relative improvements on 13 datasets over previous SOTA. Notably, it only costs 14% extra inference time compared to CLIP and without any additional training, substantially better than existing generative-model-based purification methods.
>
> Since you mentioned that there have been adversarial purification methods in uni-modal latent space for a long time, please point us to the specific paper(s), we would appreciate the opportunity to fill this knowledge gap.
>
>
> **Q3**: Please provide the revised version addressing my Q1 before the discussion close. I think the current version still is hard to read.
>
> **A3**: We have refined the abstract in the revised version to make the background more concise and easier to understand. Thank you for your valuable feedback.
>
>
> **Thank you again for your efforts. We are looking forward to your reply and are open to further discussion.**

---

> ### Comment · Reviewer_7kUG · 2024-12-01
> **Official Comment from Reviewer 7kUG**
>
> I appreciate the response from authors. I think the refined contributions are enough. Thus, I raise my rating "5->6".
>
> By the way, I still hope the authors would further improve their presentation skill (The updated version is OK). As a reviewer, I don't think it's my responsibility to provide **specific writing guidance**. I think **the seniors in authors** should take on this responsibility.

---

> > ### Author Response · Authors · 2024-12-01
> >
> > We sincerely appreciate your positive feedback and are grateful for your valuable suggestions.

---

### Official Review · Reviewer_g2Zx · 2024-11-02

**Soundness:** 3
**Presentation:** 3
**Contribution:** 3
**Rating:** 6
**Confidence:** 3

**Summary:**

This paper aims to build an adversarially robust zero-shot image classifier, based on CLIP. The authors propose two variant purification-based methods, CLIPPure-Diff and CLIPPure-Cos. The new approach is demonstrated to greatly boost the adversarial robustness and consistently set a new state-of-the-art across several datasets.

**Strengths:**

1. The paper is well motivated by measuring the KL divergence between the joint distributions of the purification and attack steps.
2. Although purification has been widely studied in pixel space, this paper showcases the potential of multi-modal latent space for learning robust zero-shot classification.
3. The proposed method is the first, as far as I know, purification method in multi-modal latent space.
4. The experiments and analysis are comprehensive and demonstrate the excellent of the proposed method

**Weaknesses:**

1. My main concern is related to the impact of VLMs. As the authors stated, the CLIP is a strong VLM and has shown superiority on many tasks. With this regard, the advantages of the proposed method could be substantiated better through an ablative study with different VLMs.
2. The used models for evaluating the proposed method are limited, compared with the prior works [1]. For example, the results on ImageNet are only based on WideResNet-50. Transformer-based models are suggested, cf.
[1] Diffusion Models for Adversarial Purification.

**Questions:**

See weakness.

**Details Of Ethics Concerns:**

No.

---

> ### Author Response · Authors · 2024-11-23
> **Reply (1/2)**
>
> **Reply (1/2)**
>
> **We appreciate the time and effort of the reviewer.** In response to the issues raised in the review, we offer the following replies:
>
> **Q1**: My main concern is related to the impact of VLMs. As the authors stated, the CLIP is a strong VLM and has shown superiority in many tasks. With this regard, the advantages of the proposed method could be substantiated better through an ablative study with different VLMs.
>
> **A1**:  Thank you for raising the point about the impact of the base model. **Our proposed method is mainly tailored for CLIP (that is why we name it CLIPure), especially CLIPure-Cos. Although purification in latent space may also be applied to other VLMs, the optimal strategy for additional VLMs is part of our future work.** We use CLIP because our objective is to develop a robust zero-shot classifier, which can efficiently defend against any unforeseen attacks. CLIP, as a popular, efficient, and effective zero-shot classifier, can achieve accurate zero-shot classification (though non-robust) and discriminative representations between benign images and adversarial examples (detailed in Section 4.1 and Section 4.2).
>
> **Despite its effectiveness in zero-shot classification, CLIP is severely vulnerable to adversarial attacks, demonstrating a robustness of 0.0% without any defense**. Even with adversarial training (e.g., TeCoA [1] and FARE [2]), its robustness remains inadequate, especially against unseen attacks, as shown in Tables 1, 2, and 4 of the original paper. Regarding clean accuracy, CLIP does not outperform specifically trained classifiers such as WideResNet-50 shown in Table 2. However, our CLIPure achieves significantly better robustness compared to adversarially-trained classifiers, e.g., AT-ConvNeXt-L [3] and AT-Swin-L [4]. CLIPure also achieves a smaller gap between robustness and clean accuracy, underscoring the superior adversarial robustness of CLIPure beyond CLIP itself.
>
> According to your valuable feedback and considering the scope of our paper, we further study the impact of different base models including different versions of CLIP and its variants, including **EVA2-CLIP, CLIPA, and CoCa**. These results are highlighted in blue in our revised paper.
>
> **Key Observations from Our Study:**
>
> - **Model Size and Latent Space**: Larger models tend to demonstrate better clean accuracy, likely leading to a lower purification risk due to better modeling of latent space, which in turn diminishes the gap between robust and clean accuracy, as visually supported in Figure 6 (Left).
>
> - **Advanced Variants**: CLIPure based on more advanced models like CLIPA exhibits a smaller gap between robustness and clean accuracy, benefiting from enhanced latent space modeling. Notably, CLIPure using the ViT-H-14 version of CLIPA as its backbone achieved an impressive 79.3% robustness on the ImageNet dataset.
>
> - **Superior Backbones** (also a response **to Q2**): Models based on the ViT backbone typically perform better than those based on ResNet, also attributed to their higher-quality latent space modeling.
> Your suggestion for an extensive ablative study involving various VLMs is invaluable and will be considered for our future research to deeply understand the influence of different VLM architectures on adversarial robustness and purification effectiveness.
>
> | Model       | Version     | Param (M) | $\quad $CLIP w/o  |   Defense   | $\qquad $CLIPure |      |
> |:--:|--|--|:--:|:--:|:--:|:--:|
> |             |             |           | Acc         | Rob  | Acc     | Rob  |
> |  | RN50        | 102       | 59.7        | 0.0  | 60.0    | 52.9 |
> |             | RN101       | 119       | 61.6        | 0.0  | 61.9    | 55.5 |
> |             | RN50x64     | 623       | 72.0        | 0.0  | 72.3    | 69.5 |
> |             | ViT-B-16    | 149       | 68.1        | 0.0  | 68.2    | 63.0 |
> |     **CLIP**           | ViT-B-32    | 151       | 62.0        | 0.0  | 62.0    | 58.1 |
> |             | ViT-L-14    | 427       | 74.9        | 0.0  | 76.3    | 72.6 |
> |             | ViT-H-14    | 986       | 77.2        | 0.0  | 77.4    | 74.4 |
> |             | ViT-big-G-14$\ $$\ $| 2539      | 80.4        | 0.0  | 80.4    | 77.6 |
> | **EVA2-CLIP** $\ $ | ViT-B-16  | 149       | 74.6        | 0.0  | 74.7    | 71.7 |
> |             | ViT-L-14    | 427       | 81.0        | 0.0  | 80.7    | 78.7 |
> | **CLIPA**   | ViT-L-14    | 414       | 79.0        | 0.0  | 79.0    | 77.2 |
> |             | ViT-H-14    | 968       | 81.8        | 0.0  | **81.5**    | **79.3** |
> | **CoCa**    | ViT-B-32    | 253       | 64.2        | 0.0  | 63.8    | 59.8 |
>
>
> **We appreciate your efforts and are open to further discussion if you have any additional concerns.**

---

> ### Author Response · Authors · 2024-11-23
> **Reply (2/2)**
>
> **Reply (2/2)**
>
> **We appreciate the time and effort of the reviewer.** In response to the issues raised in the review, we offer the following replies:
>
> **Q2**: The used models for evaluating the proposed method are limited, compared with the prior works [5]. For example, the results on ImageNet are only based on WideResNet-50. Transformer-based models are suggested, cf. [5] Diffusion Models for Adversarial Purification.
>
> **A2**: We are afraid there is a misunderstanding. **In Table 2 (results on ImageNet), WideResNet-50 is a baseline we compare with our CLIP-based classifier without defense. It is not the backbone of our CLIPure. Our CLIPure is always based on CLIP.**
>
> Unlike DiffPure, which performs purification in pixel space and is independent of the classifier, our method is built on a zero-shot classification framework, i.e., CLIP. As outlined in Eq. (2), the purified image embeddings can be directly classified in a zero-shot manner by matching images with text prompts such as "a photo of <class-name>" using CLIP. To further clarify this point, we detailed the zero-shot classification strategy in Algorithm 1 of the revised paper and included an illustration of the CLIPure process in Figure 5 of the Appendix.
>
> As detailed in the previous **response (A1) (results shown in Table 4 and Figure 6 in our revised version)**, the backbone CLIP versions of CLIPure include both **transformer-based ViT and ResNet-based models**. Results show that ViT backbones typically perform better than those based on ResNet, attributed to their higher-quality latent space modeling.
>
> [1] Understanding zero-shot adversarial robustness for large-scale models, ICLR 2023
>
> [2] Robust CLIP: Unsupervised Adversarial Fine-Tuning of Vision Embeddings for Robust Large Vision-Language Models, ICML 2024
>
> [3] Revisiting adversarial training for imagenet: Architectures, training and generalization across threat models, NeruIPS 2024
>
> [4] A comprehensive study on robustness of image classification models: Benchmarking and rethinking, IJCV 2024
>
> [5] Diffusion Models for Adversarial Purification, ICML 2022
>
> **We appreciate your efforts and are open to further discussion if you have any additional concerns.**

---

> > ### Comment · Reviewer_g2Zx · 2024-11-25
> >
> > Thanks for the detailed response and clarification. I will keep my current positive score and increase my confidence.

---

> > > ### Author Response · Authors · 2024-11-25
> > >
> > > We sincerely thank you for your positive feedback and greatly appreciate your suggestions regarding Vision Language Models (VLMs), which have inspired additional explorations. These findings further enhance the substantiation of our CLIPure. We remain open and responsive to any further discussions until the conclusion of the discussion stage.

---

### Official Review · Reviewer_k5Xb · 2024-11-03

**Soundness:** 3
**Presentation:** 3
**Contribution:** 3
**Rating:** 6
**Confidence:** 2

**Summary:**

This paper proposes CLIPure, an adversarial purification method that operates within the CLIP models' latent space. This approach can enhance adversarial robustness on zero-shot classification without additional training. Even CLIPure-Cos can improve adversarial robustness without an external diffusion model. They demonstrate the effectiveness of CLIPure on various datasets.

**Strengths:**

1. This paper robustifies CLIP in latent space instead of pixel space, and they show the strength of this strategy in analysis.

2. The proposed method can improve the adversarial robustness of zero-shot classification of CLIP without training. In particular, CLIPure-Cos even does not require generative models, which are much more efficient regarding inference cost.

**Weaknesses:**

1. Missing baselines: [1] proposed adversarial and certified robustness of CLIP, compared to TeCoA.

[1] Choi et al., Adversarial Robustification via Text-to-Image Diffusion Models, ECCV 2024

**Questions:**

Please answer the Weakness part.

---

> ### Author Response · Authors · 2024-11-22
>
> **We appreciate the time and effort of the reviewer.** In response to the issues raised in the review, we offer the following replies:
>
> **Q1**: Missing baseline "Adversarial Robustification via Text-to-Image Diffusion Models" (ECCV'24)
>
> **A1**: Thank you for pointing out the recent work "Adversarial Robustification via Text-to-Image Diffusion Models" (ECCV'24) [1]. We are previously unaware of this study since its public version was first released on July 26th, 2024 (**within three months before our submission**). Their RobustifyT2I and our CLIPure approach both use CLIP for adversarial defense, but **RobustifyT2I employs random smoothing while CLIPure is a purification method**. We conduct a comparative experiment using the same setup as RobustifyT2I with a CLIP model based on ViT-B-32 on ImageNet. Experimental results show that **CLIPure significantly outperforms RobustifyT2I in both clean and robust accuracy against an $\ell_2$ threat mode with epsilon values of {0.5, 1.0}**.
>
> |     Method     |  Robust|Accuracy (%) | $\ \ \ $ Clean|Accuracy (%)  |
> |:------------------:|:------------------:|:------------------:|:------------------:|:------------------:|
> |                    | $\epsilon=0.5$ | $\epsilon=1.0$ | $\epsilon=0.5$ | $\epsilon=1.0$ |
> | CLIP               | 1.4                 | 0.2 | 58.2                | 58.2 |
> | RobustifyT2I (w/o adapt) [1] | 40.0                | 31.0 | 56.2                | 55.2 |
> | RobustifyT2I [1]   | 42.6                | 31.4 | 57.6                | 56.2 |
> | Our **CLIPure-Cos**    | **61.3**                | **60.5** | 62.1                | 62.1 |
>
> We appreciate your valuable feedback and now include it in our related work section.
>
> [1] Choi et al., Adversarial Robustification via Text-to-Image Diffusion Models, ECCV 2024

---

> > ### Comment · Reviewer_k5Xb · 2024-11-25
> >
> > Thanks for the authors' response. The rebuttal has addressed my concerns. I will keep my positive score.

---

> > > ### Author Response · Authors · 2024-11-26
> > >
> > > We sincerely thank you for your positive feedback. We remain open and responsive to any further discussions until the end of the discussion stage.

---

### Meta-Review · Area_Chair_Qysj · 2024-12-21

**Metareview:**

This paper proposes a CLIP-based purification method, CLIPure, which includes two variants: CLIPure-Diff, which models image likelihood using a generative process on its latent vector, and CLIPure-Cos, which models likelihood based on the similarity between image embeddings and a blank template for adversarially robust zero-shot image classification. After the rebuttal, six reviewers are inclined to accept the paper. The Area Chair (AC) has carefully reviewed the feedback and discussions, finding that the rebuttal effectively addresses most of the reviewers' concerns. Therefore, the AC recommends accepting this paper.

**Additional Comments On Reviewer Discussion:**

During the rebuttal phase, the authors analyzed the impact of CLIP versions and its variant models, compared CLIPure with two recent baselines to validate its superior performance, provided more detailed clarifications of the experimental settings, included more case studies and hyperparameter analyses. After rebuttal, all reviewers tend to accept this paper. Based on these discussions, the AC agreed to accept this paper.

---

### Decision · Program_Chairs · 2025-01-22

Accept (Poster)